# Distribution-Aware Tensor Decomposition for Compression of Convolutional Neural Networks

**Alper Kalle**[*][†]
alper.kalle@cea.fr

**Theo Rudkiewicz**[*][‡]
theo.rudkiewicz@inria.fr

**Mohamed-Oumar Ouerfelli**[†]
mohamed-oumar.ouerfelli@cea.fr

**Mohamed Tamaazousti**[†]
mohamed.tamaazousti@cea.fr

## Abstract

Neural networks are widely used for image–related tasks but typically demand considerable computing power. Once a network has been trained, however, its memory- and compute-footprint can be reduced by compression. In this work, we focus on compression through tensorization and low-rank representations. Whereas classical approaches search for a low-rank approximation by minimizing an isotropic norm such as the Frobenius norm in weight-space, we use data-informed norms that measure the error in function space. Concretely, we minimize the change in the layer's output distribution, which can be expressed as $\|(W - \widetilde{W})\Sigma^{1/2}\|_F$ where $\Sigma^{1/2}$ is the square root of the covariance matrix of the layer's input and $W$, $\widetilde{W}$ are the original and compressed weights. We propose new alternating least square algorithms for the two most common tensor decompositions (Tucker-2 and CPD) that directly optimize the new norm. Unlike conventional compression pipelines, which almost always require post-compression fine-tuning, our data-informed approach often achieves competitive accuracy without any fine-tuning. We further show that the same covariance-based norm can be transferred from one dataset to another with only a minor accuracy drop, enabling compression even when the original training dataset is unavailable. Experiments on several CNN architectures (ResNet-18/50, and GoogLeNet) and datasets (ImageNet, FGVC-Aircraft, Cifar10, and Cifar100) confirm the advantages of the proposed method.

## 1 Introduction

For the past decade, neural networks have consistently outperformed other algorithms across a wide array of tasks particularly in computer vision, including character recognition, image classification, object detection, image segmentation, and depth estimation. Notably, even with the rise of other architectures, Convolutional Neural Networks (CNNs) continue to push the boundaries of performance in computer vision, with recent advancements demonstrating their sustained competitiveness for the state-of-the-art results [Liu et al., 2022, Woo et al., 2023]. A defining characteristic of these networks, in contrast to traditional algorithms (e.g., nearest neighbors, decision trees), is their substantial number of parameters. For instance, prominent architectures such as AlexNet [Krizhevsky et al., 2017], VGG-16 [Simonyan and Zisserman, 2015], ResNet [He et al., 2016] and GoogLeNet [Szegedy et al., 2015] possess tens of millions of trainable parameters.

---

[*]These authors contributed equally

[†]Université Paris-Saclay, CEA, List, F-91120, Palaiseau, France

[‡]TAU team, LISN, Université Paris-Saclay, CNRS, Inria, 91405, Orsay, France

39th Conference on Neural Information Processing Systems (NeurIPS 2025).

Despite the performance benefits often associated with a large parameter count during training, opportunities exist for creating more compact representations of trained models. This is particularly critical for applications requiring efficient execution on resource-constrained embedded devices, such as on-device image classification for smartphones and systems for autonomous vehicles. Consequently, a significant body of research has emerged, focusing on techniques to reduce the parameter count in neural networks [Cheng et al., 2018], including methods like quantization, pruning, knowledge distillation, and low-rank factorization.

Classical approaches to low-rank tensor approximation for network compression typically involve finding a compressed representation that minimizes an isotropic norm, such as the Frobenius norm, of the difference between the original and compressed weights. This minimization is performed directly in the weight space of the network. While mathematically convenient, such an approach does not explicitly consider the impact of weight changes on the network's functional behavior or its output given the data it processes. Consequently, these methods often lead to a notable drop in accuracy, necessitating a subsequent, computationally intensive fine-tuning stage to recover performance. This fine-tuning step typically requires access to the original training dataset, which may not always be available.

In this paper, we formalize a paradigm shift from classical weight-space error minimization for tensorized layers. Traditional tensorization approaches replace a computational block, typically a single layer $f_\theta$ parameterized by $\theta$, with a sequence of more compact layers—for instance, a composition $f_{A,B,C} := g_A \circ h_B \circ i_C$, where $A$, $B$, $C$ are the new, smaller parameter tensors. Classical methods then seek to find $A$, $B$, $C$ such that their effective combined parameters are close to the original $\theta$ i.e. $\|r(A, B, C) - \theta\|_F$ is small (where $r$ is the operator that combines the low rank tensors). In contrast, the data-informed approach aims to ensure that the function implemented by the sequence of compressed layers, $f_{A,B,C}$, closely approximates the original layer's function, $f_\theta$, for the actual data the network encounters. This means we measure the error directly in the function space $\|f_{A,B,C} - f_\theta\|_{L^2}$, prioritizing the preservation of the layer's input-output behavior rather than just its parameter values. This make it a functional norm and distribution-aware norm. This leads to properly justify the "data norm" that was used by Denton et al. [2014] in the case of SVD decomposition.

To operationalize this concept, we develop novel Alternating Least Squares (ALS) algorithms specifically designed to optimize this new data-informed norm for two of the most widely used tensor decomposition formats: Tucker-2 and Canonical Polyadic Decomposition (CPD). A significant advantage of our approach is its ability to achieve competitive accuracy levels often without any post-compression fine-tuning, thereby streamlining the compression pipeline and reducing its dependency on extensive retraining.

Furthermore, we demonstrate a crucial property of our covariance-based norm: its transferability. We show that the input covariance statistics learned from one dataset can be effectively applied to compress models for different, related datasets with only a minor degradation in accuracy. This finding is particularly valuable as it opens up possibilities for compressing models even when the original training data is inaccessible due to privacy, proprietary restrictions, or other constraints.

Our contributions are:

1. The use of distribution-informed norm for the neural network tensor decomposition.

2. Development of new ALS algorithms for Tucker-2 and CPD tensor decompositions that directly optimize this distribution-informed norm.

3. In depth comparison of tensor decomposition with Tucker and CP decompositions with and without the knowledge of data distribution, and also comparing two different approach: tensor deflation and CP-ALS algorithms with respect to distribution-informed norm.

4. Empirical evidence showing that our approach often achieves competitive accuracy without fine-tuning, and that the learned covariance-based norm can be transferred across datasets.

We validate our proposed methods through comprehensive experiments on several prominent CNN architectures, including ResNet-18, ResNet-50, and GoogLeNet, across diverse benchmark datasets such as ImageNet, FGVC-Aircraft, Cifar10, and Cifar100. The results consistently underscore the advantages of our distribution-informed compression strategy.

## 2 Related work

Our work intersects three active threads in neural-network compression: tensor-based methods, functional norms, and data-free compression techniques.

**Tensorization** Tensor decomposition reduces both parameter count and computational cost by approximating weight tensors with low-rank formats. Early efforts include Denton et al. [2014]; subsequent milestones span CP decomposition for CNNs [Lebedev et al., 2015], Tensor-Train for fully connected layers [Novikov et al., 2015], Tucker decomposition [Kim et al., 2016], and Tensor-Ring networks [Zhao et al., 2019]. Most of these methods follow a two-step pipeline: (i) compute a low-rank approximation, then (ii) fine-tune to restore accuracy. Several studies instead alternate tensorization with fine-tuning—e.g., the tensor-power method for successive layer compression [Astrid and Lee, 2017] and the MUSCO pipeline, which repeatedly compresses and fine-tunes [Gusak et al., 2019]. Rank selection has also been explored: variational Bayesian matrix factorization (VBMF) offers a data-free criterion, applied to Tucker layers [Kim et al., 2016] and later to CP layers [Astrid et al., 2018], building on the analytic VBMF solution of Nakajima et al. [2010].

**Functional norms** Most of the above techniques minimise error in *weight space*, which is not directly linked to a network's functional behaviour. Functional norms instead measure distances between functions. They underpin natural-gradient optimisation [Ollivier, 2017, Schwencke and Furtlehner, 2024] and can guide network growth [Verbockhaven et al., 2024]. In compression, alternative norms are less common; one example is Lohit and Jones [2022], who employ an optimal-transport loss during distillation. Another important example is Denton et al. [2014] who propose the use of two non-isotropic norms: the Mahalanobis distance and the data distance. The latest was not presented as a functional norm but, as we will show later, their data norm can in fact be seen as the $L2$ functional norm. For this data distance, they proposed replacing the approximation criterion from the Frobenius norm with a data covariance distance metric, which accounts for the empirical covariance of the input layer. They recommended using this criterion alongside either the SVD method or a greedy CP algorithm, which iteratively computes rank-one tensors for approximate CP decomposition and conducted experiments for this distance using the SVD method. Furthermore, to the best of our knowledge, we are the first to conduct multiple experiments on several classical CNNs across different datasets to assess the benefits of this new norm. Finally, we also expand the analysis to evaluate the transferability of this norm between different datasets.

**Data-free compression** Several mainstream compression techniques now operate without access to the original training data. Data-free knowledge distillation was introduced by Lopes et al. [2017]. Zero-shot quantization approaches such as ZeroQ [Cai et al., 2020] and DFQ [Nagel et al., 2019] remove the need for calibration data. For pruning, Srinivas and Babu [2015] merge redundant neurons, Tang et al. [2021] prune using synthetically generated data, and SynFlow prunes networks even before training [Tanaka et al., 2020]. To the best of our knowledge, no existing work tackles the data-free high order tensor approximation problem. We address this by evaluating the use of a generic distinct dataset to the one used during model training.

**Alternative Norms** Defining an importance score, or norm, is central to structured network compression. Prevailing methods rely on simple weight magnitude or more complex, loss-based metrics derived from the Hessian Matrix [LeCun et al., 1989] or Fisher Information Matrix [Tu et al., 2016]. Our work explores an alternative based on activation statistics. Specifically, our distribution-aware norm uses the activation covariance matrix to model the complete second-order statistical structure of feature maps. This provides a richer measure of importance than common activation-based heuristics, which are often limited to first-order statistics like mean magnitude [Rhu et al., 2018]. By capturing inter-channel dependencies, our norm offers a more holistic criterion for guiding the compression.

## 3 Background on compression by tensor decomposition

The general principle of tensor decomposition is to replace a part of the neural network by a lighter one. For example if we have a neural network $f = q \circ l \circ p$ where $q$ and $p$ are two neural networks and $l$ is one layer of the neural network, we can replace $l$ by a new squence of layers $l' := l_1 \circ l_2 \circ \cdots \circ l_n$. The new neural network is then $f' := q \circ l' \circ p$. The goal of the tensor decomposition is to find the best

$l'$ such that $f$ and $f'$ are as close as possible. In most of the case the layer $l$ is either a fully connected layer or a convolutional layer. In the case of a fully connected layer, the tensor decomposition is done through the singular value decomposition (SVD) and lead to replace the layer by a couple of fully connected layers. In the case of a convolutional layer, the tensor decomposition can be done through the CP decomposition or the Tucker decomposition as described in the next section.

**Convolution**    A convolution $\mathbf{Conv}_{\mathcal{K}}$ parameterized by a tensor $\mathcal{K}$ of size $(T, S, H, W)$ is a function from the space of images with $S$ channels like $\mathcal{X} \in \mathbb{R}^{S \times H_y \times W_x}$ where $H_y$, $W_x$ represent the height and width of the image to the space of images with $T$ channels like $\mathcal{Y} \in \mathbb{R}^{T \times H'_y \times W'_x}$. The convolution layer has $SHWT$ parameters and the cost of the convolution operation given above is $(SHWT)(H'_y W'_x)$ additions-multiplications.

By decomposing the kernel tensor we can replace the convolution by a series of smaller layers in terms parameters and computational cost. In the next sections, we present two possible decompositions that can be seen as generalizations of singular value decomposition (SVD) to the tensor case.

### 3.1    CP Decomposition for Convolutional Layer Compression

The CANDECOMP/PARAFAC (CP) decomposition [Hitchcock, 1927] represents a tensor as sum of rank 1 tensors, where the rank 1 tensors are them self decomposed as the outer product of vectors. For a rank $R$, CP decomposition of a kernel tensor $\mathcal{K} \in \mathbb{R}^{T \times S \times H \times W}$ is given by the following formula:

$$\widetilde{\mathcal{K}} = \sum_{r=1}^{R} \boldsymbol{U}_r^{(T)} \otimes \boldsymbol{U}_r^{(S)} \otimes \boldsymbol{U}_r^{(H)} \otimes \boldsymbol{U}_r^{(W)} \tag{1}$$

where $\boldsymbol{U}_r^{(n)} \in \mathbb{R}^n$ for $n \in [T, S, H, W]$ are the equivalent of the singular vectors and $\otimes$ denotes the outer product.

This decomposition leads as suggested by Lebedev et al. [2015] to replace the convolution by a series of four layers: a $1 \times 1$ convolution parameterized by $(\boldsymbol{U}_r^{(S)})_r$, a $H \times 1$ convolution parameterized by $(\boldsymbol{U}_r^{(H)})_r$, a $1 \times W$ convolution parameterized by $(\boldsymbol{U}_r^{(W)})_r$ and a $1 \times 1$ convolution parameterized by $(\boldsymbol{U}_r^{(T)})_r$. The number of parameters of the CP decomposition is $R \times (T + S + H + W)$ which for a small rank $R$ is smaller than the number of parameters of the original convolutional layer; the reduction factor is the same for the computational cost.

To choose a rank for the decompostion without reliying on trial and errors, we use the VBMF (Variational Bayesian Matrix Factorization) algorithm as suggested by Astrid et al. [2018]. We note $R_{\text{VBMF}}$ the maximum of ranks computed through application of VBMF to the each possible unfolding of the kernel tensor and $R_{\max}$ an upper bound of the rank with $R_{\max} := \frac{TSHW}{\max\{T,S,H,W\}}$. We use a rank $R_\alpha$ that is a linear combination of those ranks determined by a parameter $\alpha$ that we call VBMF ratio:

$$R_\alpha = R_{\text{VBMF}} + (1 - \alpha)(R_{\max} - R_{\text{VBMF}}). \tag{2}$$

### 3.2    Tucker Decomposition for Convolutional Layer Compression

The Tucker decomposition [Tucker, 1966] countrary to the CP decomposition use a different number of eigenvectors for each mode. In the case of convolutional layers, as the two mode corresponding to the kernel size ($H$ and $W$) are small, we do not decompose them which leads to the Tucker2 decomposition. For a kernel tensor $\mathcal{K} \in \mathbb{R}^{T \times S \times H \times W}$, the Tucker2 decomposition with rank $R_T$ and $R_S$ is $\widetilde{\mathcal{K}} = \mathcal{G} \times_1 \boldsymbol{U}^{(T)} \times_2 \boldsymbol{U}^{(S)*}$.

As suggested by Kim et al. [2016], this leads to replace the convolution by a series of two layers: a $1 \times 1$ convolution parameterized by $\boldsymbol{U}^{(T)}$, a full convolution parameterized by $\mathcal{G}$ and a $1 \times 1$ convolution parameterized by $\boldsymbol{U}^{(S)}$.

To select the ranks $R_T$ and $R_S$, we use the same method as for CP decomposition. We compute the ranks $R_{T-VBMF}$ and $R_{S-VBMF}$ by applying the VBMF algorithm to the unfolding matrices of sizes $T \times SHW$ and $S \times THW$. Then we compute the ranks $R_T$ and $R_S$ using the formula given

---

$^*\times_i$ indicates a product along the $i$th axis of the tensor $\mathcal{G}$

in equation (2) on the VBMF ranks $R_{T-VBMF}$ and $R_{S-VBMF}$, respectively (with the $R_{\max}$ being either $T$ or $S$).

# 4 The Distribution-Aware Tensor Decomposition

## 4.1 Functional Metric for Network Compression

In general when replacing the layer $l_\theta$ from $f = q \circ l_\theta \circ p$ to get $f' = q \circ l'_{\theta'} \circ p$ we don't want to retrain the network so we aim to have $f \approx f'$. Mathematically for a data distribution $\mathcal{D}$, for a classification task we want to have $\mathbb{E}_{x \sim \mathcal{D}}(d(f(x), f'(x))) \approx 0$ where $d$ is a way to measure the distance between two distributions like the Wasserstein distance or the KL divergence.

Most of the works use as proxy the Frobenius norm $\|\theta - \theta'\|_F$ where $\theta$ are the parameters of the layer $l_\theta$ and $\theta'$ are the combined parameters of the new layer $l'_{\theta'}$ (this combination is different depending on the method (see previous section)). Here we propose instead to minimize the functional norm $\|l_\theta \circ p - l'_{\theta'} \circ p\|_{\mathsf{L}^2}$ where $p$ is the network before the layer $l_\theta$ where we take as measure the data distribution $\mathcal{D}$. Hence, we have:

$$\|l_\theta \circ p - l'_{\theta'} \circ p\|_{\mathsf{L}^2} = \sqrt{\mathbb{E}_{x \sim \mathcal{D}}\left(\|l_\theta \circ p(x) - l'_{\theta'} \circ p(x)\|_F^2\right)}. \tag{3}$$

To compute the $\|\cdot\|_{\mathsf{L}^2}$ in the case where $l_\theta$ is a convolution we use the following result:

**Proposition 1.** *Consider a distribution $\mathcal{D}$, a partial neural network $p$, and two convolution $\mathbf{Conv}_{\mathcal{K}}$ and $\mathbf{Conv}_{\widetilde{\mathcal{K}}}$ parametrized by the kernel tensor $\mathcal{K} \in \mathbb{R}^{T \times S \times H \times W}$ and $\widetilde{\mathcal{K}}$. Under reasonable assumptions [†], we can define $\Sigma := \mathbb{E}_{x \sim \mathcal{D}}\left(u(p(x))u(p(x))^\top\right)$ where $u$ is the unfolding operator [‡] that transforms the image $p(x)$ into a matrix that can be used to compute the convolution as a matrix product. We can also define $\Sigma^{1/2}$ the square root of $\Sigma$ such that $\Sigma^{1/2}(\Sigma^{1/2})^\top = \Sigma$. Then, we have:*

$$\left\|\mathbf{Conv}_{\mathcal{K}} \circ p - \mathbf{Conv}_{\widetilde{\mathcal{K}}} \circ p\right\|_{\mathsf{L}^2} = \left\|\left(\mathcal{K} - \widetilde{\mathcal{K}}\right)_{(1)} \Sigma^{1/2}\right\|_F \tag{4}$$

*where $(\cdot)_{(1)}$ is the reshaping of the convolution kernel into $(T, S \times H \times W)$.*

*See appendix A for the proof.*

**Frobenius norm**    Here we can notice that in the case where $\Sigma = I_n$, for example if we consider that $\mathcal{D} = \mathcal{N}(0, I)$, we recover the Frobenius norm over the parameters of the layer. Hence the functional norm is a generalization of the Frobenius norm over the parameters of the layer, suited for the case of a non isotropic data distributions. In the following, we call the distribution-aware norm given in the Eq. 4 as Sigma norm.

## 4.2 Decomposition Computation

Although taking the input data distribution into account lead to better compression results, we cannot directly apply the highly efficient algorithms used for kernel approximation based on the Frobenius norm. In this paper, we introduce new efficient algorithms, equivalent to the widely used CP-ALS and Tucker-ALS methods for the Frobenius norm, that approximate the kernel using this new metric.

### 4.2.1 Alternating Least Square Algorithm for CP with Distribution-Aware Norm

The CP decomposition as presented in Eq. 1 involves 4 terms. The main idea of the ALS algorithm is to minimize successively each of the terms and then to iterate this process. The minimization of each term is convex and can be done with a close formula using the pseudo-inverse. We details below how the minimization can be done with Sigma norm.

---

[†]The reasonable assumptions are that the function $p$ is in $\mathsf{L}^2$ for the distribution $\mathcal{D}$. This is likely the case in our range of applications since $p$ is a neural network that is in most cases continuous and $\mathcal{D}$ can be considered as restricted to a compact set.

[‡]`https://docs.pytorch.org/docs/stable/generated/torch.nn.Unfold.html`

Firstly, we note the properties of vectorization of matrices which we will use to adopt the Sigma norm for ALS algorithm. Let $A \in \mathbb{R}^{m \times n}$ and $B \in \mathbb{R}^{n \times k}$ be two matrices. Then, we can state the followings, for all $i \in [\![1, m]\!], j \in [\![1, n]\!]$,

$$\text{Vec}(A)[m(j-1)+i] = A[i,j], \text{ and} \tag{5}$$

$$\text{Vec}(AB) = (B^\top \otimes \text{Id}(m)) \text{Vec}(A) = (\text{Id}(k) \otimes A) \text{Vec}(B). \tag{6}$$

Given that the first unfolding of CP decomposition is $\widetilde{\mathcal{K}}_{(1)} = \boldsymbol{U}^{(T)} (\boldsymbol{U}^S \odot \boldsymbol{U}^H \odot \boldsymbol{U}^W)^\top$ where $\odot$ denotes the Khatri–Rao product, we have:

$$\text{Vec}(\widetilde{\mathcal{K}}_{(1)} \Sigma^{1/2}) = \left( (\Sigma^{1/2})^\top \otimes \text{Id}(T) \right) \text{Vec}(\widetilde{\mathcal{K}}_{(1)}) \tag{7}$$

$$= \left( (\Sigma^{1/2})^\top \otimes \text{Id}(T) \right) \text{Vec}(\boldsymbol{U}^{(T)} (\boldsymbol{U}^S \odot \boldsymbol{U}^H \odot \boldsymbol{U}^W)^\top) \tag{8}$$

$$= \left( (\Sigma^{1/2})^\top \otimes \text{Id}(T) \right) \left( (\boldsymbol{U}^{(S)} \odot \boldsymbol{U}^{(H)} \odot \boldsymbol{U}^{(W)}) \otimes \text{Id}(T) \right) \text{Vec}(\boldsymbol{U}^{(T)}). \tag{9}$$

Thus, we can iterate over each factor matrices to minimize the error between the kernel tensor $\mathcal{K}$ and its approximation $\widetilde{\mathcal{K}}$ under Sigma norm. In other words, for the factor matrix $\boldsymbol{U}^{(T)}$ we have the following minimization problem:

$$\min_{U^{(T)}} \left\| \text{Vec}(\mathcal{K}\Sigma^{1/2}) - \boldsymbol{P}^{(T)} \text{Vec}(\boldsymbol{U}^{(T)}) \right\|_F, \tag{10}$$

where $\boldsymbol{P}^{(T)} = \left( (\Sigma^{1/2})^\top \otimes \text{Id}(T) \right) \left( (\boldsymbol{U}^{(S)} \odot \boldsymbol{U}^{(H)} \odot \boldsymbol{U}^{(W)}) \otimes \text{Id}(T) \right)$, and similarly for the other matrices corresponding to the components $S, H, W$, which are detailed in the appendix B.2.

Since the least square minimization problems have a closed form solution, we can deduce the factor matrix $U^{(T)}$ (and similarly the other factors) with the following formula:

$$\text{Vec}(U^{(T)}) \leftarrow (\boldsymbol{P}^{(T)})^\dagger \text{Vec}(\mathcal{K}\Sigma^{1/2}) \tag{11}$$

where $\boldsymbol{A}^\dagger$ denotes the Moore–Penrose inverse of the matrix $\boldsymbol{A}$. As a result, we present the full algorithm called CP-ALS-Sigma in algorithm 1 and we refer the appendix B.4 for the practical use of the algorithm 1.

---

**Algorithm 1** CP-ALS-Sigma

1: **function** $[\boldsymbol{U}^{(T)}, \boldsymbol{U}^{(S)}, \boldsymbol{U}^{(W)}, \boldsymbol{U}^{(H)}, m] = $ CP-ALS-SIGMA
2:      Give initializations for matrices $\boldsymbol{U}^{(T)}, \boldsymbol{U}^{(S)}, \boldsymbol{U}^{(H)}, \boldsymbol{U}^{(W)}$
3:      **for** $n = 1, \ldots, m$ **do**
4:          $\text{Vec}(\boldsymbol{U}^{(T)}) \leftarrow (\boldsymbol{P}^{(T)})^\dagger \text{Vec}(\mathcal{K}\Sigma^{1/2})$          $\triangleright$ update $\boldsymbol{U}^{(T)}$ with a least square solution
5:          $\text{Vec}(\boldsymbol{U}^{(S)}) \leftarrow (\boldsymbol{P}^{(S)})^\dagger \text{Vec}(\mathcal{K}\Sigma^{1/2})$          $\triangleright$ update $\boldsymbol{U}^{(S)}$ with a least square solution
6:          $\text{Vec}(\boldsymbol{U}^{(H)}) \leftarrow (\boldsymbol{P}^{(H)})^\dagger \text{Vec}(\mathcal{K}\Sigma^{1/2})$          $\triangleright$ update $\boldsymbol{U}^{(H)}$ with a least square solution
7:          $\text{Vec}(\boldsymbol{U}^{(W)}) \leftarrow (\boldsymbol{P}^{(W)})^\dagger \text{Vec}(\mathcal{K}\Sigma^{1/2})$          $\triangleright$ update $\boldsymbol{U}^{(W)}$ with a least square solution
8:      **return** factor matrices $\boldsymbol{U}^{(T)}, \boldsymbol{U}^{(S)}, \boldsymbol{U}^{(W)}, \boldsymbol{U}^{(H)}$

---

#### 4.2.2 Alternating Least Square Algorithm for Tucker2 with Distribution-Aware Norm

The principle of the ALS algorithm for Tucker2 is very similar to the CP case. We derive a quadratic minimization problem for each factor when both others are fixed. Those minimization problems can be solved in closed form using the matrix pseudo-inverse. In the end, this leads to the following algorithm 2. Full details can be found in appendix B.3 and the practical use of the algorithm 2 is provided in the appendix B.4.

## 5 Experimental Validation

In our experiments, we investigate the convolutional neural network models Resnet18, Resnet50, and GoogLeNet (pretrained on ImageNet 1K) to assess the performance of the proposed algorithms CP-ALS-Sigma (Algorithm 1) and Tucker2-ALS-Sigma (Algorithm 2). We do not compress the first

**Algorithm 2** Tucker2-ALS-Sigma

---
1: **function** $[\boldsymbol{U}^{(T)}, \boldsymbol{U}^{(S)}, \mathcal{G}, m] = \text{TUCKER2-ALS-SIGMA}$
2:     Give initializations for matrices $\boldsymbol{U}^{(T)}, \boldsymbol{U}^{(S)}$
3:     **for** $n = 1, \dots, m$ **do**
4:         $\text{Vec}(\boldsymbol{U}^{(T)}) \leftarrow (\boldsymbol{P}^{(T)})^{\dagger}\, \text{Vec}(\mathcal{K}\Sigma^{1/2})$          $\triangleright$ update $\boldsymbol{U}^{(T)}$ with a least square solution
5:         $\text{Vec}(\boldsymbol{U}^{(S)}) \leftarrow (\boldsymbol{P}^{(S)})^{\dagger}\, \text{Vec}(\mathcal{K}\Sigma^{1/2})$          $\triangleright$ update $\boldsymbol{U}^{(S)}$ with a least square solution
6:         $\text{Vec}(\mathcal{G}) \leftarrow (\boldsymbol{P}^{(\mathcal{G})})^{\dagger}\, \text{Vec}(\mathcal{K}\Sigma^{1/2})$          $\triangleright$ update $\mathcal{G}$ with a least square solution
7:     **return** core tensor $\mathcal{G}$ and factor matrices $\boldsymbol{U}^{(T)}, \boldsymbol{U}^{(S)}$

---

convolutional layer for each model, as doing so degrades performance. Additionally, we used the Tensorly package [Kossaifi et al., 2019] to compute the standard CP and Tucker decompositions, referred to as CP-ALS and Tucker2-ALS, respectively. Some parts of the code were also adapted from the MUSCO library [Gusak et al., 2019] to perform the compression on neural networks.

## 5.1 Evaluation of the CP-ALS and Tensor Deflation with Distribution-Aware Norm

An alternative to the full ALS method is the tensor deflation method. In details, tensor deflation method suggests to update iteratively $W^{(k+1)} \leftarrow W^{(k)} - \alpha_k \otimes \beta_k \otimes \gamma_k \otimes \delta_k$ where $\alpha_k \otimes \beta_k \otimes \gamma_k \otimes \delta_k$ is the rank-one approximation of $W^{(k)}$ such that the low rank approximation is given by $\widetilde{\mathcal{W}} = \sum_{k=1}^{R} \alpha_k \otimes \beta_k \otimes \gamma_k \otimes \delta_k$. We compare this greedy approach with our ALS algorithm that optimize all the ranks at the same time, in the case where we optimize the distribution-aware norm. In Table 1, we show the reconstruction error for the distribution-aware norm ($\left\| \left( \mathcal{K} - \widetilde{\mathcal{K}} \right)_{(1)} \Sigma^{1/2} \right\|_F \big/ \left\| \mathcal{K}_{(1)} \Sigma^{1/2} \right\|_F$

where $\mathcal{K}$ is the original tensor and $\widetilde{\mathcal{K}}$ is the decomposed one) when decomposing kernels of ResNet 18. We observe a clear superiority of the full ALS algorithm in reconstruction error (Table 1) and in accuracy after compression (Table 2).

Table 1: Relative reconstruction errors of Greedy Tensor Deflation(TD) and CP-ALS-Sigma algorithms applied to convolutional layers of ResNet18 where the ranks are estimated through VBMF with the ratio $\alpha = 0.8$.

| Conv Layer | Rank | Recons. Err. (Greedy TD) | Recons. Err. (CP-ALS-Sigma) | Improvement (Greedy / ALS) |
|---|---|---|---|---|
| Layer1.0.conv2 | 134 | 0.195 | **0.053** | 3.7 |
| Layer2.0.conv1 | 140 | 0.277 | **0.118** | 2.3 |
| Layer3.0.conv1 | 291 | 0.217 | **0.094** | 2.3 |
| Layer3.1.conv1 | 520 | 0.207 | **0.070** | 3.0 |
| Layer4.1.conv1 | 1023 | 0.159 | **0.063** | 2.5 |
| Layer4.1.conv2 | 1167 | 0.292 | **0.051** | 5.7 |

Table 2: Top-1 accuracies of the decomposed models obtained via Greedy Tensor Deflation (TD) and CP-ALS-Sigma algorithms with respect to distribution-aware norm applied to ResNet18 on ImageNet dataset with different compression rates.

| VBMF Ratio | Compression Rate | Acc. (Greedy TD) | Acc.(CP-ALS-Sigma) |
|---|---|---|---|
| 0.8 | 2.07 | 35.2 | **67.9** |
| 0.85 | 2.53 | 20.9 | **66.5** |
| 0.9 | 3.25 | 5.1 | **63.1** |

## 5.2 Model Compression and Fine-Tuning with Limited Data Access

In this section, we consider the scenario where only a limited amount of data is available, a common situation in many applications. Specifically, we consider the case where only 50,000 images from the ImageNet training set are available to estimate the matrix $\Sigma$ and to fine-tune the model.

We compare the classification accuracy of the compressed model with Tucker2-ALS-Sigma, Tucker2-ALS algorithms for Frobenius norm and its fine-tuned version. For the network compressed using the standard ALS algorithm under the Frobenius norm, we fine-tune it with the Adam optimizer, selecting the optimal learning rate from the range $10^{-5}$ to $10^{-10}$. We test the performance of our method at different compression rates by varying the parameter $\alpha$ in the rank formula (2). For this, we selected the following values of $\alpha$: $[0.4, 0.45, 0.5, 0.55]$ for Resnet18, $[0.8, 0.9, 0.95, 1]$ for Resnet50, $[0.6, 0.7, 0.8]$ for GoogLeNet.

As shown in Figure 1, our results indicate that the Tucker2-ALS-Sigma algorithm consistently outperforms both the standard Tucker2-ALS algorithm and the fine-tuned compressed models obtained with Tucker2-ALS across all the neural networks evaluated. For additional results on the CP-ALS-Sigma algorithm, we refer to Appendix C.1.

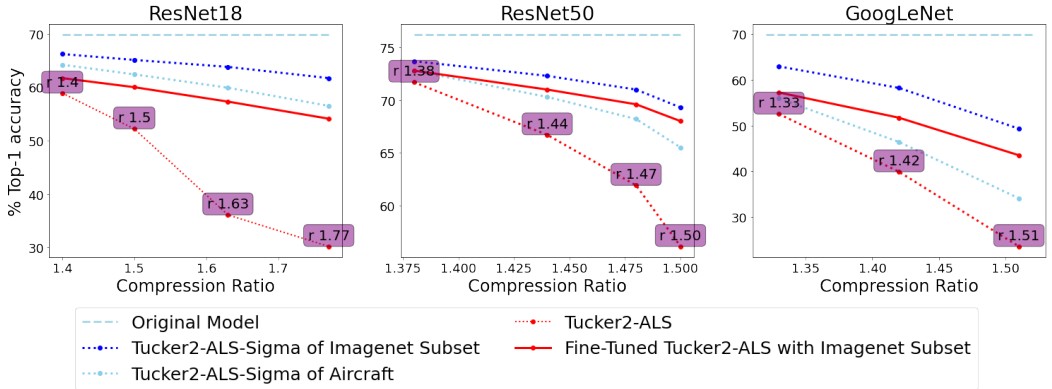

Figure 1: Accuracy comparison of decomposed models obtained with Tucker2-ALS-Sigma and Tucker2-ALS algorithms, also including fine-tuned decomposed model (with Tucker2-ALS algorithm) results where the fine-tuning done on the subset of ImageNet train dataset. ($rX$ denotes the compression ratio, calculated by dividing the number of parameters of the original model by that of the compressed model.)

## 5.3 Impact of Dataset Changes on Proposed Algorithms

This section aims to assess how variations in the dataset affect the performance of the proposed CP-ALS-Sigma and Tucker2-ALS-Sigma algorithms. In particular, we demonstrate that the distribution-aware norm can be computed from a dataset different from the one used for pretraining, highlighting its transferability. To achieve this, we performed compression on ResNet18, ResNet50, and GoogLeNet models, all trained on the CIFAR-10 dataset, using our proposed ALS-Sigma algorithms (with the $\Sigma$ matrix obtained from different datasets) and the standard ALS algorithm, applied to both CP and Tucker decompositions. In addition, we fine-tuned the compressed models that are obtained using the standard ALS algorithms to compare their performance with the Sigma-based methods (without fine-tuning). Fine-tuning was conducted on the CIFAR-10 training dataset, employing the Adam optimizer and testing various learning rates from $10^{-3}$ to $10^{-7}$, selecting the best-performing learning rate. Results for the CP-ALS-Sigma algorithm can be found in Appendix C.2.

We compare Tucker2-ALS-Sigma with the standard Tucker2-ALS algorithm and the fine-tuned compressed model obtained from Tucker2-ALS. We chose $\alpha$ from $[0.5, 0.6, 0.7, 0.8, 0.9, 1]$ for Resnet18, $[0.9, 1, 1.1, 1.2, 1.3, 1.4]$ for Resnet50, $[0.6, 0.7, 0.8, 0.9, 0.95, 1]$ for GoogLeNet. Our results, shown in Figure 2, indicate that Tucker2-ALS-Sigma algorithm has better performance than the standard Tucker2-ALS algorithm across all models, even when the $\Sigma$ matrix is computed using different datasets, such as a subset of the ImageNet training set or the CIFAR-100 training set. Additionally, while Tucker2-ALS experiences rapid performance degradation as the compression rate increases, our method remains much more consistent. Furthermore, the Tucker2-ALS-Sigma algorithm produces results that are close to those of the fine-tuned compressed model obtained using the Tucker2-ALS algorithm. Notably, the performance of the Tucker2-ALS-Sigma algorithm on the CIFAR-100 dataset is comparable to that on the CIFAR-10 dataset, suggesting that our approach is not limited to using the original training dataset for compression with a distribution-aware norm. In

fact, as illustrated in Figure 2, the results for the Tucker2-ALS-Sigma algorithm with the CIFAR-100 dataset even outperform those with CIFAR-10 for the ResNet50 model. However, when the Sigma matrix is derived from a subset of the ImageNet training set, the performance of the Tucker2-ALS-Sigma algorithm is somewhat less effective compared to when it is computed on CIFAR-10 or CIFAR-100, likely due to differences in image resolution. We refer to the Appendix G for additional experiments which investigate the effects of image resolution, dataset diversity, and the number of samples chosen for the $\Sigma$ matrix on the performance of proposed algorithms.

Figure 1 demonstrates that the Tucker2-ALS-Sigma algorithm, when the Sigma matrix is derived from the FGVC-Aircraft training dataset, yields higher classification accuracy than the standard Tucker2-ALS method for all tested ImageNet models. Notably, in the case of ResNet18, the Tucker2-ALS-Sigma variant surpasses even the fine-tuned model obtained with the standard algorithm.

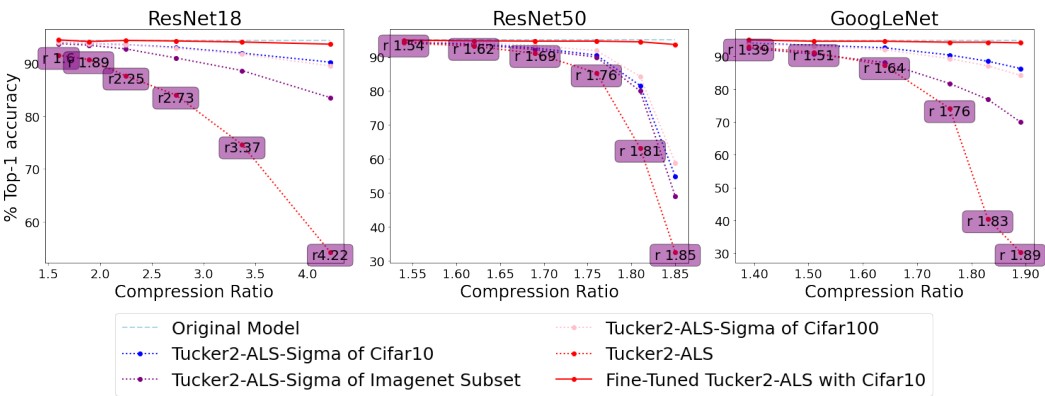

Figure 2: Accuracy comparison of Tucker2-ALS-Sigma and Tucker2-ALS algorithms including the fine-tuned model results after compression with Tucker2-ALS using Cifar10 dataset.

## 5.4 Evaluation Against Alternative Data-Informed Baselines

To rigorously evaluate our approach, we designed and implemented two strong, data-aware baseline methods inspired by metrics from pruning and activation analysis: Fisher information [Tu et al., 2016] and activation sparsity [Rhu et al., 2018]. Each baseline guides the Tucker2 decomposition by solving a specific Weighted Alternating Least Squares (WALS) problem, enabling a direct comparison against our proposed method.

**Fisher-Weighted Low-Rank Approximation (FW-LRA)** This baseline adapts Fisher information, a well-established criterion from pruning literature, to guide decomposition. The method solves the weighted objective $\min ||\mathbf{H}_{\text{fisher}} \circledast (\mathcal{K} - \widetilde{\mathcal{K}})||_F^2$, where the weighting tensor $\mathbf{H}_{\text{fisher}}$ is derived from the diagonal of the Fisher Information Matrix. The intuition is to prioritize the preservation of weights that are most sensitive to the network's loss function.

**Activation-Guided Low-Rank Approximation (AG-LRA)** Inspired by sparsity-based compression techniques, this baseline prioritizes filters based on their activation magnitude. It solves the objective $\min ||\mathbf{H}_{\text{act}} \circledast (\mathcal{K} - \widetilde{\mathcal{K}})||_F^2$, where the weighting tensor $\mathbf{H}_{\text{act}}$ is calculated from the mean absolute activation of each output channel. This heuristic assumes that filters with higher average activation are more critical for the network's predictions.

**Comparison with Baselines** Tables 3 and 4 summarize the accuracy of our Tucker2-ALS-Sigma method compared to the baseline Tucker2-ALS and the two data-aware variants on GoogLeNet and ResNet18, respectively. Across all compression rates, our method consistently outperforms both FW-LRA and AG-LRA, as well as the vanilla baseline Tucker2-ALS, demonstrating the effectiveness of our distribution-aware norm.

Table 3: Accuracy comparison of different data-aware baseline methods and Tucker2-ALS-Sigma with Tucker2-ALS at various compression rates on GoogLeNet.

| Compr. Rate | Tucker2-ALS | AG-LRA | FW-LRA | Our Method |
|---|---|---|---|---|
| 1.24 | 60.2 | 60.5 | 53.0 | **66.3** |
| 1.33 | 52.5 | 54.9 | 35.8 | **64.2** |
| 1.42 | 39.9 | 43.0 | 6.0 | **60.1** |
| 1.51 | 23.6 | 25.1 | 1.1 | **52.2** |

Table 4: Accuracy comparison of different data-aware baseline methods and Tucker2-ALS-Sigma with Tucker2-ALS at various compression rates on Resnet18.

| Compr. Rate | Tucker2-ALS | AG-LRA | FW-LRA | Our Method |
|---|---|---|---|---|
| 1.4 | 58.9 | 54.9 | 36.7 | **66.8** |
| 1.5 | 52.2 | 45.0 | 18.9 | **66.1** |
| 1.63 | 36.1 | 37.5 | 12.2 | **64.9** |
| 1.77 | 30.2 | 31.2 | 4.2 | **63.3** |

## 6 Limitations and Future Work

**Complete functional norm** A first limitation of our work is that when minimizing the reconstruction error in the layer $l$ we only take into account the first part of the network in the functional norm. Indeed, we want to minimize $\|f - f'\|$ where $f = q \circ l_\theta \circ p$ and $f' = q \circ l'_{\theta'} \circ p$. To do so we minimize $\|l_\theta \circ p - l'_{\theta'} \circ p\|_{L^2}$ as a proxy. This is an important improvement compared to the standard proxy $\|\theta - \theta'\|_F$ but this could be improved by taking the part $q$ of the network into account in the optimization process.

**Anisotropic VBMF** To choose the rank of the decomposed tensors we use the Variational Bayesian Matrix Factorization (VBMF) algorithm. This algorithm is based on the assumption that the matrix we try to factorize are perturbed by a Gaussian isotropic noise. Similarly to replacing the Frobenius norm by the functional norm, we could use a more general assumption on the noise. Doing so would require to design a new algorithm to compute the VBMF rank but could lead to a better rank choice.

**Performance guarantee** In data-free compression, we show that our algorithm improves performance for all targeted compression ratios. However, we are not able to give a performance guarantee of the compressed network without testing it on a dataset. Hence to select the rank we rely on a priori heuristic like the VBMF rank. We believe that a possible future work would be to use the reconstruction error in the functional norm to select the rank. Indeed, preliminary experiments shown in the appendix K demonstrate that the reconstruction error in the functional norm is a way better indicator of the performance of the compressed network than the reconstruction error in the Frobenius norm.

## 7 Conclusion

In summary, we have shown that incorporating *distribution-aware* (functional) norms into tensor-based network compression leads to substantial performance gains. By deriving ALS procedures that directly optimize CP and Tucker decompositions under the Sigma norm, we achieve markedly lower reconstruction error and higher accuracy than with traditional Frobenius-based methods. In addition, CP-ALS-Sigma consistently surpasses greedy tensor deflation optimized with the same norm. The advantage of our approach becomes even more pronounced at higher compression rates, where standard methods degrade sharply. Remarkably, the distribution-aware decompositions recover almost all of the accuracy otherwise obtained by fine-tuning models compressed with conventional ALS, yet they require no additional training when the $\Sigma$ matrix is estimated from the data distribution. Even when that distribution is learned from a smaller, dataset such as FGVC-Aircraft, the benefits persist. Finally, when the original training data are unavailable, the $\Sigma$ matrix can be transferred from a related dataset, still delivering significant improvements—highlighting the practicality and robustness of our distribution-aware compression framework.

# 8 Acknowledgements

We thank Clément Laroudie and Charles Villard for helpful discussions that contributed to this work. This publication was made possible by the use of the FactoryIA supercomputer, financially supported by the Ile-de-France Regional Council. This work is also supported by the PEPR-IA : ANR-23-PEIA-0010 and DeepGreen : ANR-23-DEGR-0001.

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

# A  Proof on functional norm

**Proposition 1.** *Consider a distribution $\mathcal{D}$, a partial neural network $p$, and two convolution $\mathbf{Conv}_{\mathcal{K}}$ and $\mathbf{Conv}_{\widetilde{\mathcal{K}}}$ parametrized by the kernel tensor $\mathcal{K} \in \mathbb{R}^{T \times S \times H \times W}$ and $\widetilde{\mathcal{K}}$. Under reasonable assumptions [§], we can define $\Sigma := \mathbb{E}_{x \sim \mathcal{D}}\big(u(p(x))u(p(x))^{\top}\big)$ where $u$ is the unfolding operator [¶] that transforms the image $p(x)$ into a matrix that can be used to compute the convolution as a matrix product. We can also define $\Sigma^{1/2}$ the square root of $\Sigma$ such that $\Sigma^{1/2}\big(\Sigma^{1/2}\big)^{\top} = \Sigma$. Then, we have:*

$$\left\|\mathbf{Conv}_{\mathcal{K}} \circ p - \mathbf{Conv}_{\widetilde{\mathcal{K}}} \circ p\right\|_{\mathsf{L}^2} = \left\|\left(\mathcal{K} - \widetilde{\mathcal{K}}\right)_{(1)} \Sigma^{1/2}\right\|_{F} \tag{12}$$

*Proof.  Convolution as a matrix product*

We can write the convolution as a matrix product. We first need to unfold the image $p(x)$ of size $S \times H_y \times W_x$ into a matrix $u(p(x))$ of size $(S \times H \times W, H'_y \times W'_x)$. In this matrix each line is a patch of the image $p(x)$ of size $(S \times H \times W)$. We can also reshape the kernel $\mathcal{K}$ of size $(T, S, H, W)$ into a matrix $\mathcal{K}_{(1)}$ of size $(T, S \times H \times W)$. The convolution can then be written as:

$$\mathbf{Conv}_{\mathcal{K}} \circ p(x) = \mathcal{K}_{(1)} u(p(x)) \in (T, H'_y \times W'_x). \tag{13}$$

*Technical compuations* Let $x$ an input data, we have:

$$\left\|\mathbf{Conv}_{\mathcal{K}} \circ p(x) - \mathbf{Conv}_{\widetilde{\mathcal{K}}} \circ p(x)\right\|_{F}^{2} = \left\|\mathcal{K}_{(1)}u(p(x)) - \widetilde{\mathcal{K}}_{(1)}u(p(x))\right\|_{F}^{2} \tag{14}$$

$$= \left\|\left(\mathcal{K} - \widetilde{\mathcal{K}}\right)_{(1)} u(p(x))\right\|_{F}^{2} \tag{15}$$

$$= \left\langle \left(\mathcal{K} - \widetilde{\mathcal{K}}\right)_{(1)} u(p(x)), \left(\mathcal{K} - \widetilde{\mathcal{K}}\right)_{(1)} u(p(x)) \right\rangle. \tag{16}$$

Using the fact that $\langle u, vw \rangle = \langle uw^{\top}, v \rangle$, we have:

$$= \left\langle \left(\mathcal{K} - \widetilde{\mathcal{K}}\right)_{(1)} u(p(x))u(p(x))^{\top}, \left(\mathcal{K} - \widetilde{\mathcal{K}}\right)_{(1)} \right\rangle. \tag{17}$$

*Taking the expectation* Now we take the expectation over the data distribution $\mathcal{D}$:

$$\left\|\mathbf{Conv}_{\mathcal{K}} \circ p - \mathbf{Conv}_{\widetilde{\mathcal{K}}} \circ p\right\|_{\mathsf{L}^2}^{2} = \mathbb{E}_{x \sim \mathcal{D}}\left(\left\|\mathbf{Conv}_{\mathcal{K}} \circ p(x) - \mathbf{Conv}_{\widetilde{\mathcal{K}}} \circ p(x)\right\|_{F}^{2}\right). \tag{18}$$

By linearity of the expectation and of the matrix multiplication and using (17), we have:

$$= \left\langle \left(\mathcal{K} - \widetilde{\mathcal{K}}\right)_{(1)} \mathbb{E}_{x \sim \mathcal{D}}\big(u(p(x))u(p(x))^{\top}\big), \left(\mathcal{K} - \widetilde{\mathcal{K}}\right)_{(1)} \right\rangle \tag{19}$$

$$= \left\langle \left(\mathcal{K} - \widetilde{\mathcal{K}}\right)_{(1)} \Sigma, \left(\mathcal{K} - \widetilde{\mathcal{K}}\right)_{(1)} \right\rangle \tag{20}$$

$$= \left\langle \left(\mathcal{K} - \widetilde{\mathcal{K}}\right)_{(1)} \Sigma^{1/2}\big(\Sigma^{1/2}\big)^{\top}, \left(\mathcal{K} - \widetilde{\mathcal{K}}\right)_{(1)} \right\rangle. \tag{21}$$

Using the fact that $\langle uw^{\top}, v \rangle = \langle u, vw \rangle$ we have:

$$= \left\langle \left(\mathcal{K} - \widetilde{\mathcal{K}}\right)_{(1)} \Sigma^{1/2}, \left(\mathcal{K} - \widetilde{\mathcal{K}}\right)_{(1)} \Sigma^{1/2} \right\rangle \tag{22}$$

$$= \left\|\left(\mathcal{K} - \widetilde{\mathcal{K}}\right)_{(1)} \Sigma^{1/2}\right\|_{F}^{2}. \tag{23}$$

---

[§]The reasonable assumptions are that the function $p$ is in $\mathsf{L}^2$ for the distribution $\mathcal{D}$. This is likely the case in our range of applications since $p$ is a neural network that is in most cases continuous and $\mathcal{D}$ can be considered as restricted to a compact set.

[¶]`https://docs.pytorch.org/docs/stable/generated/torch.nn.Unfold.html`

Hence, we have:

$$\left\|\mathbf{Conv}_{\mathcal{K}} \circ p - \mathbf{Conv}_{\widetilde{\mathcal{K}}} \circ p\right\|_{\mathsf{L}^2} = \left\|\left(\mathcal{K} - \widetilde{\mathcal{K}}\right)_{(1)} \Sigma^{1/2}\right\|_F. \tag{24}$$

$\square$

## B   Details on ALS Algorithm with Distribution-Aware Norm

### B.1   Computation of the Square Root of $\Sigma$

#### B.1.1   Definition and Existence

As we stated in the Proposition 1, we have $\Sigma := \mathbb{E}_{x \sim \mathcal{D}}\big(u(p(x))u(p(x))^\top\big)$. In this section, we show that the square root of the matrix $\Sigma$ can be computed via its Cholesky decomposition. In this end, we need to show that the matrix $\mathbb{E}_{x \sim \mathcal{D}}\big(u(p(x))u(p(x))^\top\big)$ is a symmetric positive definite matrix.

First of all, note that $\Sigma$ is symmetric:

$$\Sigma^\top = \big[\mathbb{E}_{x \sim \mathcal{D}}\big(u(p(x))u(p(x))^\top\big)\big]^\top = \mathbb{E}_{x \sim \mathcal{D}}\big([u(p(x))u(p(x))^\top]^\top\big) \tag{25}$$

$$= \mathbb{E}_{x \sim \mathcal{D}}\big(u(p(x))u(p(x))^\top\big) = \Sigma. \tag{26}$$

Moreover, $\Sigma$ is a positive semi-definite matrix, since for any vector $z \in \mathbb{R}^n$, we have

$$z^\top \Sigma z = z^\top \mathbb{E}_{x \sim \mathcal{D}}\big(u(p(x))u(p(x))^\top\big)z \tag{27}$$

$$= \mathbb{E}_{x \sim \mathcal{D}}\big(z^\top u(p(x))u(p(x))^\top z\big) = \mathbb{E}_{x \sim \mathcal{D}}\left(\left\|z^\top u(p(x))\right\|_F^2\right) \geq 0. \tag{28}$$

To guarantee that $\Sigma$ admits a Cholesky decomposition, it must be positive definite. This requires that the columns of the matrix $u(p(x))$ are linearly independent and span the space in which they lie.

Thus, the matrix $\Sigma$ admits a Cholesky decomposition if and only if it is positive definite, which holds when the covariance of $u(p(x))$ is full rank. However, in practice, the columns of $u(p(x))$ can be linearly dependent, making $\Sigma$ only positive semi-definite. To ensure numerical stability and enable the Cholesky decomposition, we add a small regularization term $\epsilon \mathrm{Id}$, where $\epsilon > 0$ and $\mathrm{Id}$ is the identity matrix, to the diagonal of $\Sigma$ in cases where it is only positive semi-definite. Alternatively, the square root of $\Sigma$ can be computed using the singular value decomposition (SVD), which remains applicable even when $\Sigma$ is not full rank.

#### B.1.2   Complexity Analysis

**Computation of $\Sigma$**   Let recall the notations, for the case where we want to compute $\Sigma$ at layer $l$ of a network:

- $N$ the number of samples used to estimate $\Sigma$
- $p$ the function corresponding to the first part of the network computing the layer $l$
- $C$ the complexity of one forward pass of the full network
- $H_y[l+1], W_x[l+1]$ the height and width of layer $l+1$
- $S$ the number of channels of layer $l$
- $H, W$ the height and width of the kernel of the connection from layer $l$ to layer $l+1$

To compute $\Sigma$ for each $x$ we compute:

- $p(x)$ which has a complexity bounded by $C$
- For each of the $H_y[l+1]W_x[l+1]$ patches $p_{i,j}$ of size $S \times H \times W$ in $p(x)$ (each patch corresponds to the receptive field of a neuron of layer $l+1$) we compute $(p_{i,j})^\top p_{i,j}$ which has a complexity of $(SHW)^2$ per patch. Then we sum all those matrices to get $u(p(x))\,(u(p(x)))^\top$. The complexity of this step is $H_y[l+1]W_x[l+1](SHW)^2$.

In total, the complexity of the computation of $\Sigma$ is $O(NC + NH_y[l+1]W_x[l+1](SHW)^2)$. In comparison, the complexity of forwarding trought the connection from layer $l$ to layer $l+1$ is $O(H_y[l+1]W_x[l+1]STHW)$ where $T$ is the number of channels of layer $l+1$. Thus the computation of $\Sigma$ is at most $\frac{SHW}{T}$ times the complexity of a classical forward (whith typical values $S \approx T$ and $H, W$ small, the computation of $\Sigma$ is comparable to the complexity of a few (9 for $3 \times 3$ kernels and 49 for the biggest $7 \times 7$ kernels) forward passes).

**Computation of the Square Root of** $\Sigma$    Computing the square root of $\Sigma$ can be done with Cholesky or SVD which require $O((SHW)^3)$ operations in both cases. This cost is independant of $N$ thus as long as $N > SHW$ the cost of computing $\Sigma$ is higher than the cost of computing its square root.

**Conclusion**    In general, the computation of $\Sigma$ and its square root is comparable to at most a few (9 for $3 \times 3$ kernels and 49 for the biggest $7 \times 7$ kernels) forward passes through the network times the number of samples used to estimate $\Sigma$.

## B.2    Details of the CP-ALS-Sigma Algorithm

Remember that for a given rank $R$, CP decomposition of a kernel tensor $\mathcal{K} \in \mathbb{R}^{T \times S \times H \times W}$ is given by the following formula:

$$\widetilde{\mathcal{K}} = \sum_{r=1}^{R} \boldsymbol{U}_r^{(T)} \otimes \boldsymbol{U}_r^{(S)} \otimes \boldsymbol{U}_r^{(H)} \otimes \boldsymbol{U}_r^{(W)}. \tag{29}$$

By iterating over the components of the CP decomposition as given in the Section 4.2.1, we obtain the following sequence of minimization problems:

$$\min_{\boldsymbol{U}^{(T)}} \left\| \mathrm{Vec}(\mathcal{K}\Sigma^{1/2}) - \underbrace{\left( (\Sigma^{1/2})^\top (\boldsymbol{U}^{(S)} \odot \boldsymbol{U}^{(H)} \odot \boldsymbol{U}^{(W)}) \otimes \mathrm{Id}(T) \right)}_{\boldsymbol{P}^{(T)}} \mathrm{Vec}(\boldsymbol{U}^{(T)}) \right\|_F, \tag{30}$$

$$\min_{\boldsymbol{U}^{(S)}} \left\| \mathrm{Vec}(\mathcal{K}\Sigma^{1/2}) - \underbrace{((\Sigma^{1/2})^\top \otimes \mathrm{Id}(T)) \left[ (\boldsymbol{U}^{(T)} \odot \boldsymbol{U}^{(W)} \odot \boldsymbol{U}^{(H)}) \otimes \mathrm{Id}(S) \right]}_{=: \boldsymbol{P}^{(S)}} \mathrm{Vec}(\boldsymbol{U}^{(S)}) \right\|_F, \tag{31}$$

$$\min_{\boldsymbol{U}^{(H)}} \left\| \mathrm{Vec}(\mathcal{K}\Sigma^{1/2}) - \underbrace{((\Sigma^{1/2})^\top \otimes \mathrm{Id}(T)) \left[ (\boldsymbol{U}^{(T)} \odot \boldsymbol{U}^{(S)} \odot \boldsymbol{U}^{(W)}) \otimes \mathrm{Id}(H) \right]}_{=: \boldsymbol{P}^{(H)}} \mathrm{Vec}(\boldsymbol{U}^{(H)}) \right\|_F, \tag{32}$$

$$\min_{\boldsymbol{U}^{(W)}} \left\| \mathrm{Vec}(\mathcal{K}\Sigma^{1/2}) - \underbrace{((\Sigma^{1/2})^\top \otimes \mathrm{Id}(T)) \left[ (\boldsymbol{U}^{(T)} \odot \boldsymbol{U}^{(S)} \odot \boldsymbol{U}^{(H)}) \otimes \mathrm{Id}(W) \right]}_{=: \boldsymbol{P}^{(W)}} \mathrm{Vec}(\boldsymbol{U}^{(W)}) \right\|_F, \tag{33}$$

where we used the property $(\boldsymbol{A} \otimes \boldsymbol{C})(\boldsymbol{B} \otimes \boldsymbol{C}) = (\boldsymbol{A}\boldsymbol{B} \otimes \boldsymbol{C})$ for (30). In consequence, the associated algorithm called CP-ALS-Sigma is described in the algorithm 1.

## B.3 Details of the Tucker2-ALS-Sigma Algorithm

Remember that Tucker2 decomposition of $\mathcal{K} \in \mathbb{R}^{T \times S \times H \times W}$ is given by a core tensor $\mathcal{G} \in \mathbb{R}^{R_T \times R_S \times H \times W}$ and two factor matrices $\boldsymbol{U}^{(T)} \in \mathbb{R}^{T \times R_T}$ and $\boldsymbol{U}^{(S)} \in \mathbb{R}^{S \times R_S}$ such that

$$\widetilde{\mathcal{K}} = \mathcal{G} \times_2 \boldsymbol{U}^{(S)} \times_1 \boldsymbol{U}^{(T)}.^{\parallel}$$

Similar to CP decomposition above, using the properties of tensor unfolding and vectorization of matrices, we end up with the following minimization problems:

$$\min_{\boldsymbol{U}^{(T)}} \left\| \mathrm{Vec}(\mathcal{K}\Sigma^{1/2}) - \underbrace{\left[ \left( (\Sigma^{1/2})^\top (\mathcal{G} \times_2 \boldsymbol{U}^{(S)})_{(1)}^\top \right) \otimes \mathrm{Id}(T) \right]}_{=:\boldsymbol{P}^{(T)}} \mathrm{Vec}(\boldsymbol{U}^{(T)}) \right\|_F , \tag{34}$$

$$\min_{\boldsymbol{U}^{(S)}} \left\| \mathrm{Vec}(\mathcal{K}\Sigma^{1/2}) - \underbrace{\left( (\Sigma^{1/2})^\top \otimes \mathrm{Id}(T) \right) \left[ (\mathcal{G} \times_1 \boldsymbol{U}^{(T)})_{(2)}^\top \otimes \mathrm{Id}(S) \right]}_{=:\boldsymbol{P}^{(S)}} \mathrm{Vec}(\boldsymbol{U}^{(S)}) \right\|_F , \tag{35}$$

and

$$\min_{\mathcal{G}} \left\| \mathrm{Vec}(\mathcal{K}\Sigma^{1/2}) - \underbrace{\left( (\Sigma^{1/2})^\top \otimes \mathrm{Id}(T) \right) \left[ \boldsymbol{U}^{(T)} \otimes \boldsymbol{U}^{(S)} \otimes \mathrm{Id}(H) \otimes \mathrm{Id}(W) \right]}_{=:\boldsymbol{P}^{(\mathcal{G})}} \mathrm{Vec}(\mathcal{G}) \right\|_F . \tag{36}$$

Thus, we associate the corresponding factor matrix $\boldsymbol{U}^{(T)}$ (and similarly for the factor matrix $\boldsymbol{U}^{(S)}$ and core tensor $\mathcal{G}$) with

$$\mathrm{Vec}(\boldsymbol{U}^{(T)}) \leftarrow (\boldsymbol{P}^{(T)})^\dagger \mathrm{Vec}(\mathcal{K}\Sigma^{1/2}). \tag{37}$$

Consequently, the full algorithm Tucker2-ALS-Sigma is presented in Algorithm 2.

## B.4 Pseudo inverse computation for CP-ALS-Sigma and Tucker2-ALS-Sigma Algorithms

We refer the section 4.2.1 implementing Sigma norm on the CP-ALS algorithm, we obtain the closed form solution for the factor matrix $\boldsymbol{U}^{(T)}$ (and similarly for the other factors) with the following formula:

$$\mathrm{Vec}(\boldsymbol{U}^{(T)}) \leftarrow (\boldsymbol{P}^{(T)})^\dagger \mathrm{Vec}(\mathcal{K}\Sigma^{1/2}). \tag{38}$$

Since $\boldsymbol{P}^{(T)}$ is a matrix of size $TSHW \times RT$, we apply the identity $\boldsymbol{A}^\dagger = (\boldsymbol{A}^\top \boldsymbol{A})^\dagger \boldsymbol{A}^\top$ to avoid explicitly forming and storing large intermediate matrices of size $TSHW \times RT$, $TSHW \times RS$, $TSHW \times RH$, and $TSHW \times RW$, which would otherwise be required for computing the factor matrices $\boldsymbol{U}^{(T)}$, $\boldsymbol{U}^{(S)}$, $\boldsymbol{U}^{(H)}$, and $\boldsymbol{U}^{(W)}$, respectively.

Then, we can rewrite the above equation (38) as:

$$\mathrm{Vec}(\boldsymbol{U}^{(T)}) \leftarrow \left( (\boldsymbol{P}^{(T)})^\top \boldsymbol{P}^{(T)} \right)^\dagger (\boldsymbol{P}^{(T)})^\top \mathrm{Vec}(\mathcal{K}\Sigma^{1/2}), \tag{39}$$

such that $(\boldsymbol{P}^{(T)})^\top \mathrm{Vec}(\mathcal{K}\Sigma^{1/2})$ is a vector of size $RT$ and $(\boldsymbol{P}^{(T)})^\top \boldsymbol{P}^{(T)}$ is a matrix of size $RT \times RT$. However, in practice, we observed that the matrix $(\boldsymbol{P}^{(T)})^\top \boldsymbol{P}^{(T)}$ can be ill-conditioned, making the computation of its pseudoinverse numerically unstable. To mitigate this issue, we reformulate the problem as a linear system and solve it using the MINRES (Minimum Residual) method, which is particularly effective for solving symmetric but large-scale systems. Specifically, we define

$$b = (\boldsymbol{P}^{(T)})^\top \mathrm{Vec}(\mathcal{K}\Sigma^{1/2}), \quad A = (\boldsymbol{P}^{(T)})^\top \boldsymbol{P}^{(T)},$$

and solve the system

$$A\boldsymbol{U} = b,$$

---

$^{\parallel} \times_i$ indicates a product along the $i$th axis of the tensor $\mathcal{G}$

where the solution vector $U$ corresponds to the desired factor matrix. This system is solved efficiently on the GPU using the `minres` function provided by the CuPy library.

Likewise to CP decomposition, we solve the factor matrices $U^{(T)}$ and $U^{(S)}$, and the core tensor $\mathcal{G}$ of the Tucker2 decomposition using the the `minres` function provided by the CuPy library.

## C   Experiments with CP-ALS-Sigma Algorithm

### C.1   Model Compression and Fine-Tuning with Limited Data Access

Refering the Section 5.2, we consider the case when only a limited amount of data is available, a common situation in many applications. Specifically, we consider the case where only 50,000 images from the ImageNet training set are available to estimate the matrix $\Sigma$ and to fine-tune the model.

Here, we conducted the compression algorithms on ResNet18 and GoogLeNet, and we compare the classification accuracy of the compressed model with CP-ALS-Sigma, CP-ALS algorithms for Frobenius norm and its fine-tuned version. For the network compressed using the standard ALS algorithm under the Frobenius norm, we fine-tune it with the Adam optimizer, selecting the optimal learning rate from the range $10^{-5}$ to $10^{-10}$. In addition, we selected the following values of $\alpha$: $[0.8, 0.85, 0.9]$ for Resnet18, and $[0.85, 0.9, 0.95]$ for GoogLeNet.

According to the results on ResNet18, presented in Figure 3, the CP-ALS-Sigma algorithm consistently outperforms both the standard CP-ALS algorithm and the fine-tuned compressed models obtained with CP-ALS. In addition, our experiments on GoogLeNet (see Figure 3) show that the CP-ALS-Sigma algorithm achieves higher accuracy than the standard CP-ALS while yielding close results to the fine-tuned compressed model obtained via CP-ALS.

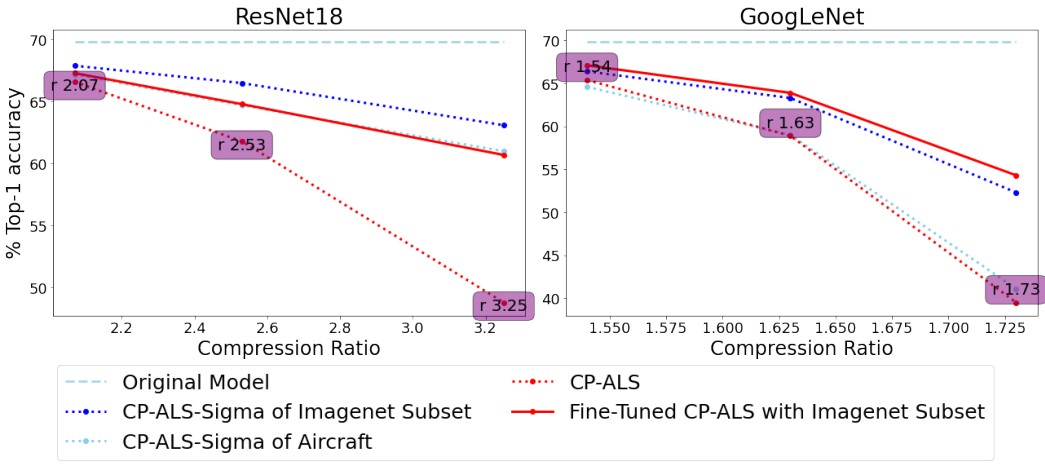

Figure 3: Accuracy comparison of decomposed models obtained with CP-ALS-Sigma and CP-ALS algorithms, also including fine-tuned decomposed model (with CP-ALS algorithm) results where fine-tuning done on the subset of ImageNet train dataset.

### C.2   Impact of Dataset Changes on Proposed Algorithms

Here, we provide the detailed results for the CP-ALS-Sigma algorithm in the dataset transferability experiments described in Section 5.3. Specifically, we evaluate the compression performance of CP-ALS-Sigma on ResNet18, and GoogLeNet models trained on CIFAR-10, including $\Sigma$ matrices derived from datasets different than the original training set. Our goal is to assess how the choice of dataset for computing the distribution-aware norm affects the final accuracy of the compressed models.

We compare CP-ALS-Sigma against the standard CP-ALS algorithm, and the fine-tuned compressed model obtained with CP-ALS. Fine-tuning was conducted on the CIFAR-10 training set using the Adam optimizer and a range of learning rates from $10^{-3}$ to $10^{-7}$, selecting the best performing

rate per model. In addition, we selected parameters $\alpha$ from $[0.85, 0.9, 0.95, 1]$ for Resnet18, and $[0.9, 0.95, 1]$ for GoogLeNet.

The results shown in Figure 4 indicate that the CP-ALS-Sigma algorithm consistently outperforms the standard CP-ALS algorithm across all models, even when the $\Sigma$ matrix is computed from different datasets, such as a subset of the ImageNet training set or the CIFAR-100 training set. Additionally, as for Tucker2-ALS experiences, while CP-ALS experiences rapid performance degradation as the compression rate increases, our method remains much more consistent. Notably, the CP-ALS-Sigma algorithm achieves similar performance on the CIFAR-100 dataset as it does on CIFAR-10, indicating that the use of a distribution-aware norm for compression is not restricted to the original training dataset. However, when the Sigma matrix is derived from a subset of the ImageNet training set, the performance of the CP-ALS-Sigma algorithm is somewhat less effective compared to when it is computed on CIFAR-10 or CIFAR-100, due to differences in image resolution as observed in Tucker2-ALS-Sigma experiments, see the appendix G for further investigation of dataset features that affect algorithm performance. Moreover, as illustrated in Figure 4, using the CP-ALS-Sigma algorithm on ResNet18 results in only a 0.8% accuracy drop at a compression rate of 3.53, and a 1.7% drop at a compression rate of 5.22, relative to the original model accuracy of 94.3% where $\Sigma$ is computed using the CIFAR-10 train dataset.

Furthermore, as illustrated in Figure 3, the compressed model obtained using the CP-ALS-Sigma algorithm, with the Sigma matrix computed from the FGVC-Aircraft training dataset, achieves close classification accuracy to the fine-tuned compressed model obtained through the standard CP-ALS algorithm applied to the original ResNet18.

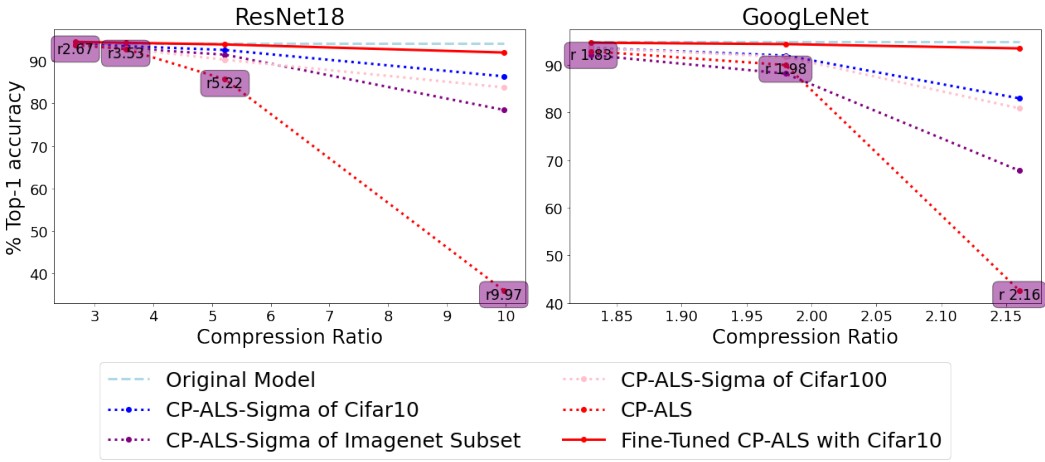

Figure 4: Comparison of CP-ALS-Sigma and CP-ALS algorithms including with fine-tuned model after compression with CP-ALS using CIFAR-10 dataset.

## D  Additional Experiments on the Impact of Dataset Changes Using CIFAR-100 Trained Models

Building on our previous experiments with CIFAR-10 trained neural network architectures (see the Section 5.3), we continue to assess how variations in the dataset influence compression outcomes and further demonstrate that the distribution-aware norm retains its transferability—even when computed from a dataset different from the one used for pretraining. Specifically, we compress ResNet18, ResNet50, and GoogLeNet models trained on CIFAR-100 using our ALS-Sigma methods—with the $\Sigma$ matrix derived from datasets different from the original train dataset—and compare the results against those of the standard ALS approach under both CP and Tucker decomposition. In addition, we fine-tuned the compressed models that are obtained using the standard ALS algorithms to compare their performance with the Sigma-based methods (without fine-tuning). Fine-tuning was performed on the CIFAR-100 training dataset, employing the Adam optimizer and testing various learning rates from $10^{-3}$ to $10^{-7}$, selecting the best-performing learning rate.

**Tucker2-ALS-Sigma** Firstly, we compare the results of Tucker2-ALS-Sigma with those of the standard Tucker2-ALS algorithm and fine-tuned compressed model obtained with Tucker2-ALS algorithm. We selected $\alpha$ from $[0.4, 0.5, 0.6, 0.7, 0.8, 0.9, 1]$ for Resnet18, $[0.9, 1, 1.1, 1.2, 1.3]$ for Resnet50, $[0.6, 0.7, 0.8, 0.9, 0.95, 1]$ for GoogLeNet. The results, presented in Figure 5, confirm the earlier observations: the Tucker2-ALS-Sigma algorithm consistently outperforms the standard Tucker2-ALS method across all evaluated models. The observed performance difference remains evident even when the $\Sigma$ matrix is derived from datasets other than the one used for training, such as a subset of ImageNet or the CIFAR-10 dataset.

Moreover, we verified again that while Tucker2-ALS causes to significant accuracy degradation as compression rate increases, the Tucker2-ALS-Sigma algorithm maintains significantly more consistent performance. Importantly, Tucker2-ALS-Sigma remains comparable in performance to fine-tuned Tucker2-ALS models—despite not involving any additional fine-tuning. These results reaffirm the transferability and effectiveness of our distribution-aware norm implementation under varying dataset conditions.

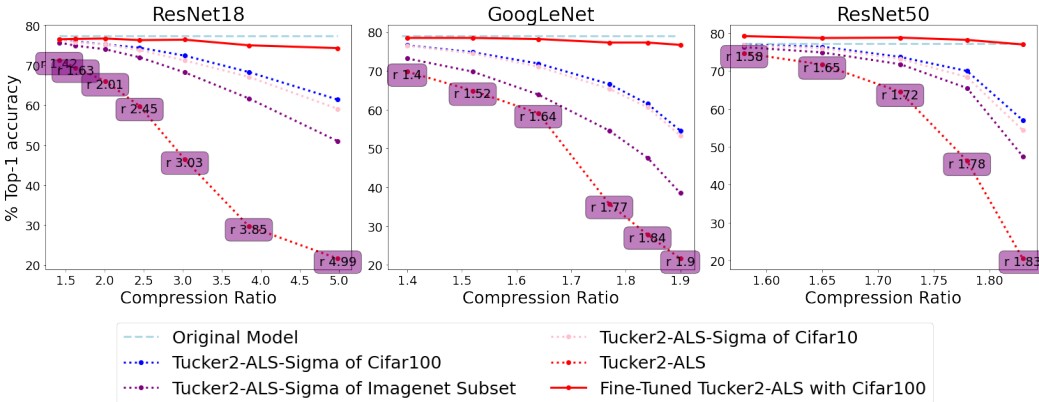

Figure 5: Accuracy comparison of decomposed models obtained using the Tucker2-ALS-Sigma and Tucker2-ALS algorithms across ResNet18, GoogLeNet, and ResNet50 architectures. The results also include fine-tuned decomposed models obtained via Tucker2-ALS, where fine-tuning was performed on the CIFAR-100 training dataset.

**CP-ALS-Sigma** Here, we present an accuracy comparison of decomposed models obtained using CP-ALS-Sigma, and the standard CP-ALS algorithm, and the fine-tuned compressed model obtained with CP-ALS. We selected $\alpha$ from $[0.85, 0.9, 0.95, 1]$ both for Resnet18, and GoogLeNet. The results shown in Figure 4 indicate that the CP-ALS-Sigma algorithm consistently outperforms the standard CP-ALS algorithm across all models, even when the $\Sigma$ matrix is computed from different datasets, such as a subset of the ImageNet training set or the CIFAR-10 training set. Additionally, as for the Tucker2-ALS experiences, while CP-ALS experiences rapid performance degradation as the compression rate increases, our method remains much more consistent. Notably, we verified again that since the CP-ALS-Sigma algorithm achieves similar performance on the CIFAR-10 dataset as it does on CIFAR-100, the use of a distribution-aware norm for compression is not restricted to the original training dataset.

In addition, when the Sigma matrix is derived from a subset of the ImageNet training set, the performance of the ALS-Sigma algorithms are somewhat less effective compared to when it is computed on CIFAR-10 or CIFAR-100, likely due to differences in image resolution as observed in experiments performed on the CIFAR-10 trained models (see the Sections 5.3 and C.2).

# E   Additional Experiments on AlexNet

In this section, we evaluate our proposed algorithms on the AlexNet architecture, extending the compression to the fully-connected layers using Singular Value Decomposition (SVD).

First, we introduce **SVD-Sigma**, a method that adapts SVD to minimize error with respect to the Sigma norm. The goal is to find a low-rank approximation $\widetilde{\mathcal{W}}$ for a given weight matrix $\mathcal{W}$ by solving

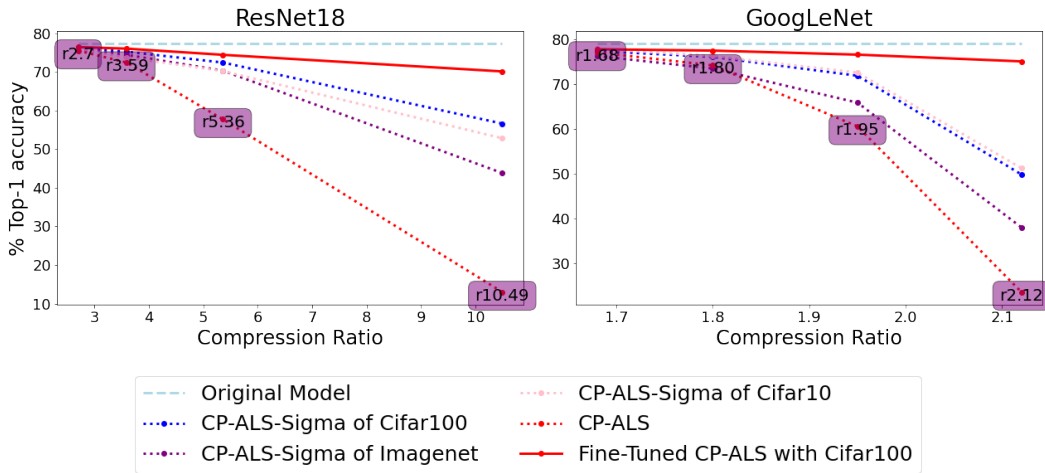

Figure 6: Accuracy comparison of decomposed models obtained using the CP-ALS-Sigma and CP-ALS algorithms across ResNet18 and GoogLeNet architectures. The results also include fine-tuned decomposed models obtained via CP-ALS, where fine-tuning was performed on the CIFAR-100 training dataset.

the following minimization problem:

$$\min_{\widetilde{\mathcal{W}}} \|(\mathcal{W} - \widetilde{\mathcal{W}})\Sigma^{1/2}\|_F. \tag{40}$$

We then compare the performance of two hybrid compression schemes in Table 5:

1. **Baseline Hybrid:** The convolutional layers are compressed with the standard **Tucker2-ALS** algorithm, while the linear layers are compressed with conventional **SVD**.

2. **Proposed Hybrid:** The convolutional layers are compressed with our **Tucker2-ALS-Sigma** algorithm, while the linear layers are compressed with the proposed **SVD-Sigma** method.

Table 5: Top-1 accuracies of the compressed AlexNet architecture using Tucker2-ALS followed by SVD, and Tucker2-ALS-Sigma followed by SVD-Sigma, where the Sigma matrix is computed using 5,000 images from the ImageNet training dataset. AlexNet original Top-1 accuracy is % 56.

| VBMF Ratio | Compression Rate | Tucker2-ALS-Sigma + SVD-Sigma | Tucker2-ALS + SVD |
|---|---|---|---|
| 0.6 | 1.46 | 49.8 | 42.4 |
| 0.65 | 1.63 | 48.1 | 40.1 |
| 0.7 | 1.85 | 46.1 | 34.4 |
| 0.8 | 2.54 | 40.9 | 25.8 |

## F   Stability of the Tensor Decomposition Methods

We examine the stability of our proposed CP-ALS-Sigma algorithm relative to the standard CP-ALS baseline. Experiments are performed on GoogLeNet trained on ImageNet, CIFAR-10, and CIFAR-100, each evaluated across multiple random seeds to measure the sensitivity of the decompositions to initialization. As shown in Tables 6, 7 and 8, CP-ALS is more sensitive to random initialization compared to CP-ALS-Sigma, particularly under high compression rates.

For Tucker2-ALS and Tucker2-ALS-Sigma, we adopt SVD-based initialization, which yields deterministic and thus fully reproducible results.

Table 6: Performance of CP-ALS and CP-ALS-Sigma on GoogLeNet (ImageNet) with the $\Sigma$ matrix computed with a subset of ImageNet training dataset. Top-1 accuracy results are reported as mean $\pm$ standard deviation, obtained using 5 different random seeds.

| VBMF Ratio | Compr. Rate | CP-ALS | $\Sigma$ of 50k-ImageNet |
|---|---|---|---|
| 0.85 | 1.54 | $65.85 \pm 0.45$ | $66.4 \pm 0.07$ |
| 0.9 | 1.63 | $59.5 \pm 0.58$ | $63.25 \pm 0.05$ |
| 0.95 | 1.73 | $37.43 \pm 2$ | $52.45 \pm 0.12$ |

Table 7: Performance of CP-ALS and CP-ALS-Sigma on GoogLeNet (trained on CIFAR-10) with the $\Sigma$ matrix computed with CIFAR-10, CIFAR-100, and a subset of ImageNet training datasets. Top-1 accuracy results are reported as mean $\pm$ standard deviation, obtained using 5 different random seeds.

| VBMF Ratio | Compr. Rate | CP-ALS | $\Sigma$ of 5k-ImageNet | $\Sigma$ of CIFAR-100 | $\Sigma$ of CIFAR-10 |
|---|---|---|---|---|---|
| 0.9 | 1.54 | $93.06 \pm 0.15$ | $92.58 \pm 0.12$ | $93.44 \pm 0.09$ | $93.53 \pm 0.08$ |
| 0.95 | 1.63 | $89.14 \pm 0.62$ | $88.31 \pm 0.24$ | $91.47 \pm 0.06$ | $91.88 \pm 0.04$ |
| 1 | 1.73 | $37.71 \pm 3.0$ | $69.86 \pm 0.31$ | $79.94 \pm 0.24$ | $82.22 \pm 0.5$ |

Table 8: Performance of CP-ALS and CP-ALS-Sigma on GoogLeNet (trained on CIFAR-100) with the $\Sigma$ matrix computed with CIFAR-10, CIFAR-100, and a subset of ImageNet training datasets. Top-1 accuracy results are reported as mean $\pm$ standard deviation, obtained using 5 different random seeds.

| VBMF Ratio | Compr. Rate | CP-ALS | $\Sigma$ of 5k-ImageNet | $\Sigma$ of CIFAR-10 | $\Sigma$ of CIFAR-100 |
|---|---|---|---|---|---|
| 0.85 | 1.68 | $76.95 \pm 0.32$ | $77.1 \pm 0.08$ | $77.7 \pm 0.19$ | $77.64 \pm 0.19$ |
| 0.9 | 1.80 | $73.46 \pm 0.48$ | $74.97 \pm 0.11$ | $76.41 \pm 0.15$ | $76.36 \pm 0.10$ |
| 0.95 | 1.95 | $61.4 \pm 1.22$ | $67.3 \pm 0.38$ | $72.54 \pm 0.09$ | $72.4 \pm 0.17$ |
| 1 | 2.12 | $21.25 \pm 1.84$ | $38.59 \pm 0.45$ | $51.28 \pm 0.11$ | $50.74 \pm 0.17$ |

## G   Impact of Dataset Properties on Algorithm Performance

### G.1   Image Resolution

This section investigates the hypothesis that differences in image resolution are a primary factor in cross-dataset performance degradation. Our experiments confirm that algorithm performance is sensitive to resolution matching and the quality of the image scaling process.

First, we found that matching the proxy dataset's resolution (ImageNet) to the target dataset's resolution (CIFAR-10) via downsampling leads to a significant performance recovery. The results in Tables 9 and 10 show that using a $\Sigma$ matrix computed on downsampled ImageNet images consistently outperforms one computed on original-resolution images.

Second, our method is sensitive to the choice of interpolation used during preprocessing. As shown in Table 11, replacing the default bilinear interpolation (used in other experiments) with bicubic interpolation during the image resizing operation yields better results, underscoring the importance of preserving feature quality during upsampling.

### G.2   Dataset Diversity

This section evaluates the impact of dataset diversity on the performance of our proposed algorithms. Our findings indicate that the method is remarkably flexible regarding the specific semantic composition of the data used to compute the $\Sigma$ matrix.

First, we found that broad dataset diversity is beneficial but not essential. The results in Table 12 confirm that a large-scale dataset like ImageNet yields the best performance. Nevertheless, performance remains strong even with smaller, more specialized datasets like FGVC-Aircraft or Oxford Flowers-102.

Table 9: Accuracy comparison of the Tucker2-ALS-Sigma algorithm using Sigma matrices computed from different datasets on a ResNet-18 model trained with CIFAR-10. The Sigma matrices were derived from the full CIFAR-10 and CIFAR-100 training sets, as well as 50,000 ImageNet training images. For the ImageNet (Down) and Aircraft (Down) variants, images were first downsampled to CIFAR-10 resolution ($3 \times 32 \times 32$) and then upsampled to ImageNet resolution ($3 \times 224 \times 224$) before computing the Sigma matrix, whereas for the other variants, images were only upsampled to ImageNet resolution.

| VBMF Ratio | Compr. Rate | $\Sigma$ of CIFAR-10 | $\Sigma$ of CIFAR-100 | $\Sigma$ of ImageNet (Orig.) | $\Sigma$ of ImageNet (Down) | $\Sigma$ of Aircraft (Orig.) | $\Sigma$ of Aircraft (Down) |
|---|---|---|---|---|---|---|---|
| 0.5 | 1.6 | 93.9 | 93.9 | 93.7 | 94 | 93.1 | 93.6 |
| 0.6 | 1.89 | 93.7 | 93.7 | 93.4 | 93.7 | 92.1 | 93.3 |
| 0.7 | 2.25 | 93.5 | 93.5 | 92.7 | 93.2 | 89.8 | 92.4 |
| 0.8 | 2.73 | 93 | 92.8 | 91 | 92.5 | 86.9 | 91.1 |
| 0.9 | 3.37 | 91.9 | 91.7 | 88.6 | 91 | 81.7 | 88.6 |
| 1 | 4.22 | 90.2 | 89.5 | 83.5 | 87.9 | 72 | 82.7 |

Table 10: Accuracy comparison of the Tucker2-ALS-Sigma algorithm using Sigma matrices computed from different datasets on a GoogleNet model trained with CIFAR-10. The Sigma matrices were derived from the full CIFAR-10 and CIFAR-100 training sets, as well as 5000 ImageNet training images.

| VBMF Ratio | Compr. Rate | $\Sigma$ of CIFAR-10 | $\Sigma$ of CIFAR-100 | $\Sigma$ of 5k-ImageNet (Orig.) | $\Sigma$ of 5k-ImageNet (Down) |
|---|---|---|---|---|---|
| 0.6 | 1.39 | 94 | 93.8 | 92.5 | 93.4 |
| 0.7 | 1.51 | 93.4 | 93.1 | 90.7 | 92.3 |
| 0.8 | 1.64 | 92.6 | 92 | 88.1 | 90.4 |
| 0.9 | 1.76 | 90.4 | 89.3 | 81.7 | 87 |
| 0.95 | 1.83 | 88.5 | 87.1 | 76.9 | 83.3 |
| 1 | 1.89 | 86.3 | 84.2 | 70 | 78.6 |

Table 11: Accuracy comparison of the Tucker2-ALS-Sigma algorithm showing the impact of upsampling quality when using low-resolution datasets for the computation of Sigma matrix on ResNet18 trained with ImageNet. The Sigma matrices were derived from the full CIFAR-10 and CIFAR-100 training sets.

| VBMF Ratio | Compr. Rate | $\Sigma$ of CIFAR-10 (Bilinear) | $\Sigma$ of CIFAR-10 (Bicubic) | $\Sigma$ of CIFAR-100 (Bilinear) | $\Sigma$ of CIFAR-100 (Bicubic) |
|---|---|---|---|---|---|
| 0.4 | 1.4 | 55.4 | 56.3 | 55.6 | 56.7 |
| 0.45 | 1.5 | 52.3 | 53.8 | 52.4 | 53.9 |
| 0.5 | 1.63 | 47.2 | 50.5 | 48.3 | 50.8 |
| 0.55 | 1.77 | 41.7 | 46.5 | 42 | 47 |

Furthermore, we investigated whether performance is sensitive to the specific theme of the proxy dataset. As detailed in Table 13, computing $\Sigma$ matrix from a single broad category (e.g., 'dogs', 'reptiles', or 'aircraft') or from mix of diverse categories has minor impact on the performance, highlighting the method's robustness to the semantic composition of the proxy data.

Table 12: Accuracy comparison of the Tucker2-ALS-Sigma algorithm using Sigma matrices computed from different datasets on the original ResNet18 architecture. The Sigma matrices were derived from the FGVC-Aircraft and Oxford Flowers-102 training sets, and the 50.000 images of ImageNet.

| VBMF Ratio | Compr. Rate | $\Sigma$ of 50k-ImageNet | $\Sigma$ of Aircraft | $\Sigma$ of Flowers-102 |
|---|---|---|---|---|
| 0.4 | 1.4 | 66.3 | 64.2 | 64.6 |
| 0.45 | 1.5 | 65.2 | 62.4 | 63.2 |
| 0.5 | 1.63 | 63.9 | 59.9 | 61.6 |
| 0.55 | 1.77 | 61.9 | 56.5 | 59.1 |

Table 13: Accuracy comparison of the Tucker2-ALS-Sigma algorithm using Sigma matrices computed depending on the categories of ImageNet dataset on the original ResNet18 architecture. We selected the categories Aircraft, Dog, and Reptiles from ImageNet to compute the Sigma matrices and called by ImageNet-Aircraft, ImageNet-Dog, and ImageNet-Reptiles. For ImageNet-MixCateg, we have chosen images from 10 different categories of ImageNet.

| VBMF Ratio | Compr. Rate | $\Sigma$ of ImageNet-Aircraft | $\Sigma$ of ImageNet-Dog | $\Sigma$ of ImageNet-Reptiles | $\Sigma$ of ImageNet-MixCateg |
|---|---|---|---|---|---|
| 0.4 | 1.4 | 65.6 | 66.1 | 65.3 | 66.6 |
| 0.45 | 1.5 | 64.3 | 65 | 64.2 | 65.7 |
| 0.5 | 1.63 | 62.6 | 63.7 | 62.3 | 64.4 |
| 0.55 | 1.77 | 60 | 61.7 | 60.2 | 62.6 |

## G.3 Sample Size

We next investigate the sensitivity of our method to the number of samples used to compute the $\Sigma$ matrix. The results, presented in Tables 14 and 15, indicate that performance is remarkably stable across different sample sizes. This robustness is a key practical advantage, as it shows our algorithm does not require an excessively large dataset for effective covariance estimation.

Table 14: Performance of Tucker2-ALS-Sigma on ResNet-18 with the $\Sigma$ matrix computed using varying sample sizes. Top-1 accuracy results are reported as mean $\pm$ standard deviation, with each sample subset chosen randomly over 5 different seeds.

| VBMF Ratio | Compr. Rate | $\Sigma$ of 5k-ImageNet | $\Sigma$ of 10k-ImageNet | $\Sigma$ of 20k-ImageNet | $\Sigma$ of 50k-ImageNet |
|---|---|---|---|---|---|
| 0.4 | 1.4 | $66.81 \pm 0.02$ | $66.80 \pm 0.03$ | $66.80 \pm 0.02$ | $66.80 \pm 0.02$ |
| 0.45 | 1.5 | $66.09 \pm 0.03$ | $66.11 \pm 0.06$ | $66.10 \pm 0.02$ | $66.08 \pm 0.02$ |
| 0.5 | 1.63 | $64.88 \pm 0.03$ | $64.90 \pm 0.05$ | $64.91 \pm 0.02$ | $64.89 \pm 0.01$ |
| 0.55 | 1.77 | $63.25 \pm 0.05$ | $63.29 \pm 0.07$ | $63.31 \pm 0.03$ | $63.30 \pm 0.01$ |

Table 15: Performance of Tucker2-ALS-Sigma on GoogLeNet with the $\Sigma$ matrix computed using varying sample sizes. Top-1 accuracy results are reported as mean $\pm$ standard deviation, with each sample subset chosen randomly over 5 different seeds.

| VBMF Ratio | Compr. Rate | $\Sigma$ of 5k-ImageNet | $\Sigma$ of 10k-ImageNet | $\Sigma$ of 20k-ImageNet | $\Sigma$ of 50k-ImageNet |
|---|---|---|---|---|---|
| 0.6 | 1.33 | $64.17 \pm 0.04$ | $64.13 \pm 0.05$ | $64.16 \pm 0.07$ | $64.17 \pm 0.03$ |
| 0.7 | 1.42 | $60.04 \pm 0.03$ | $60.11 \pm 0.06$ | $60.13 \pm 0.04$ | $60.11 \pm 0.04$ |
| 0.8 | 1.51 | $52.21 \pm 0.09$ | $52.18 \pm 0.08$ | $52.20 \pm 0.08$ | $52.17 \pm 0.05$ |

# H    Synergy of Tensor Decomposition and Quantization

Tensor decomposition and quantization are two principal and complementary model compression techniques. While our factorization approach reduces model size by exploiting structural redundancy, quantization enhances efficiency by reducing numerical precision (e.g., to INT8) of model weights. Since these methods target different aspects of compression, combining them is expected to provide additive benefits, as shown in previous work [Gui et al., 2019].

To validate this idea further, we applied post-training FP16 and INT8 quantization to our compressed GoogLeNet (trained on CIFAR-10) models. The results in Table 16 confirm this strong synergy, particularly for our **Tucker2-ALS-Sigma** algorithm. Across all compression ratios, applying FP16 quantization to the factorized model results in a negligible change in accuracy. Even with more aggressive INT8 quantization, the performance remains remarkably high, demonstrating that our method is robust to a subsequent reduction in precision.

Table 16: Top-1 accuracies of GoogLeNet(trained on CIFAR-10) compressed with **Tucker2-ALS-Sigma** and **Tucker2-ALS**. The table compares the performance without quantization (FP32) against post-training quantization (FP16 and INT8) at various compression rates.

| Compr. Rate | Tucker2-ALS-Sigma | | | Tucker2-ALS | | |
| | WO Quant. | W Quant. | | WO Quant. | W Quant. | |
| | FP32 | FP16 | INT8 | FP32 | FP16 | INT8 |
| --- | --- | --- | --- | --- | --- | --- |
| 1.39 | 93.97 | 93.94 | 93.65 | 92.81 | 92.82 | 92.47 |
| 1.51 | 93.43 | 93.39 | 93.19 | 91.16 | 91.18 | 91.24 |
| 1.64 | 92.69 | 92.68 | 92.29 | 87.66 | 87.68 | 88.38 |
| 1.76 | 90.65 | 90.68 | 90.02 | 76.43 | 76.34 | 77.03 |
| 1.83 | 88.77 | 88.79 | 88.19 | 43.57 | 43.56 | 44.31 |
| 1.89 | 86.73 | 86.73 | 85.91 | 33.73 | 33.71 | 34.33 |

While Tucker decomposition is generally robust to post-factorization quantization, the high sensitivity of Canonical Polyadic (CP) decomposition can lead to significant accuracy degradation. To address this challenge, Cherniuk et al. [2024] recently proposed a quantization-aware framework called ADMM-EPC. Their method uses a novel CP-EPC initialization to produce low-rank factors that are inherently robust to quantization, enabling the successful combination of CP decomposition and numerical precision reduction.

# I    Compute Resources

To ensure reproducibility, we detail the computational resources used for all experiments:

- **Hardware:** All CP-ALS and Tucker2-ALS experiments were executed exclusively on CPU, using a machine with an `x86_64` architecture and 1.5TB of RAM. In contrast, all CP-ALS-Sigma and Tucker2-ALS-Sigma experiments were performed on systems equipped with either an NVIDIA A100 GPU (80GB) or an NVIDIA H100 GPU (100GB), alongside the same CPU and memory configuration.

- **Software Environment:** We used PyTorch 2.6.0 and CUDA 12.4. Experiments were run on a Linux system with Python 3.10.14.

- **Execution Time:** The runtime of Tucker2 and CP decompositions varied depending on model size and compression ratio. Fine-tuning phases were run for 30 epochs, taking approximately 1–6 hours per model.

# J    Extended Background on Compression by Tensor Decomposition

We detail the implementation of CP and Tucker decompositions for convolutional layers, as introduced in Section 3. We recall the convolution $\mathbf{Conv}_{\mathcal{K}}$ parameterized by a tensor $\mathcal{K}$ of size $(T, S, H, W)$,

which defines a mapping from the space of images $\mathcal{X} \in \mathbb{R}^{S \times H_y \times W_x}$ to the space of images $\mathcal{Y} \in \mathbb{R}^{T \times H_y' \times W_x'}$. Assuming stride 1 and no padding, the output of the convolution at a given spatial location can be written as

$$\mathcal{Y}[t, y, x] = \sum_{s=1}^{S} \sum_{h=-h_d}^{h_d} \sum_{w=-w_d}^{w_d} \mathcal{K}[t, s, h, w] \mathcal{X}[s, y + h, x + w] \tag{41}$$

where $2h_d + 1 = H$ and $2w_d + 1 = W$, and the output dimensions are reduced accordingly: $H_y' = H_y - H + 1$ and $W_x' = W_x - W + 1$. For a detailed introduction to the CP and Tucker decomposition methods, we refer to [Lebedev et al., 2015]. We reproduce some of the key equations from that paper below.

## J.1 CP Decomposition for Convolutional Layer Compression

Recall that for a rank $R$, CP decomposition of a kernel tensor $\mathcal{K} \in \mathbb{R}^{T \times S \times H \times W}$ is given by the following formula:

$$\widetilde{\mathcal{K}} = \sum_{r=1}^{R} \boldsymbol{U}_r^{(T)} \otimes \boldsymbol{U}_r^{(S)} \otimes \boldsymbol{U}_r^{(H)} \otimes \boldsymbol{U}_r^{(W)} \approx \mathcal{K} \tag{42}$$

such that it can be written as:

$$\widetilde{\mathcal{K}}[t, s, h, w] = \sum_{r=1}^{R} \boldsymbol{U}_r^{(T)}[t] \boldsymbol{U}_r^{(W)}[w] \boldsymbol{U}_r^{(H)}[h] \boldsymbol{U}_r^{(S)}[s]. \tag{43}$$

As proposed by Lebedev et al. [2015], we replace the original kernel tensor $\mathcal{K}$ with its CP decomposition $\widetilde{\mathcal{K}}$, thereby expressing the convolutional layer as a sequence of 4 successive layers we obtain the followings:

$$\mathcal{Y}[t, y, x] = \sum_{r=1}^{R} \sum_{w=-w_d}^{w_d} \sum_{h=-h_d}^{h_d} \sum_{s=1}^{S} U_r^{(T)}[t] U_r^{(W)}[w] U_r^{(H)}[h] U_r^{(S)}[s] \mathcal{X}[s, y + h, x + w] \tag{44}$$

$$= \sum_{r=1}^{R} U_r^{(T)}[t] \underbrace{\left[ \sum_{w=-w_d}^{w_d} U_r^{(W)}[w] \underbrace{\left[ \sum_{h=-h_d}^{h_d} U_r^{(H)}[h] \underbrace{\left[ \underbrace{\sum_{s=1}^{S} U_r^{(S)}[s] \mathcal{X}[s, y + h, x + w]}_{1 \times 1 \text{ conv}} \right]}_{\text{depthwise conv}} \right]}_{\text{depthwise conv}} \right]}_{1 \times 1 \text{ convolution}}.$$

$$\tag{45}$$

## J.2 Tucker Decomposition for Convolutional Layer Compression

Remember that given the ranks $R_T$ and $R_S$ corresponding to the mode-1 and mode-2 unfoldings (i.e., the first and second axes) of the kernel tensor $\mathcal{K} \in \mathbb{R}^{T \times S \times H \times W}$, the Tucker2 decomposition of $\mathcal{K}$ is

$$\widetilde{\mathcal{K}} = \mathcal{G} \times_1 \boldsymbol{U}^{(T)} \times_2 \boldsymbol{U}^{(S)}. \tag{46}$$

Accordingly, the elementwise representation of the decomposed kernel at a given spatial location is given by:

$$\widetilde{\mathcal{K}}[t, s, h, w] = \sum_{r_s=1}^{R_S} \sum_{r_t=1}^{R_T} \mathcal{G}[r_t, r_s, h, w] \boldsymbol{U}^{(T)}[r_t, t] \boldsymbol{U}^{(S)}[r_s, s]. \tag{47}$$

Similarly, as suggested by Kim et al. [2016], this leads to replace the convolution by a series of the three layers: a $1 \times 1$ convolution parameterized by $\boldsymbol{U}^{(T)}$, a full convolution parameterized by $\mathcal{G}$ and a $1 \times 1$ convolution parameterized by $\boldsymbol{U}^{(S)}$. Specifically, this can be expressed as:

$$\mathcal{Y}[t,y,x] = \sum_{s=1}^{S} \sum_{h=-h_d}^{h_d} \sum_{w=-w_d}^{w_d} \sum_{r_s=1}^{R_S} \sum_{r_t=1}^{R_T} \mathcal{G}[r_t, r_s, h, w] \boldsymbol{U}^{(T)}[r_t, t] \boldsymbol{U}^{(S)}[r_s, s] \mathcal{X}[s, y+h, x+w]$$

$$= \underbrace{\sum_{r_t=1}^{R_T} \boldsymbol{U}^{(T)}[t, r_t] \left[ \underbrace{\sum_{h=-h_d}^{h_d} \sum_{w=-w_d}^{w_d} \sum_{r_s=1}^{R_S} \mathcal{G}[r_t, r_s, h, w] \left[ \underbrace{\sum_{s=1}^{S} \boldsymbol{U}^{(S)}[s, r_s] \mathcal{X}[s, h+y, w+x]}_{1 \times 1 \text{ conv}} \right]}_{H \times W \text{ conv}} \right]}_{1 \times 1 \text{ conv}}.$$

## K    Correlation Between the Reconstruction Error and Accuracy

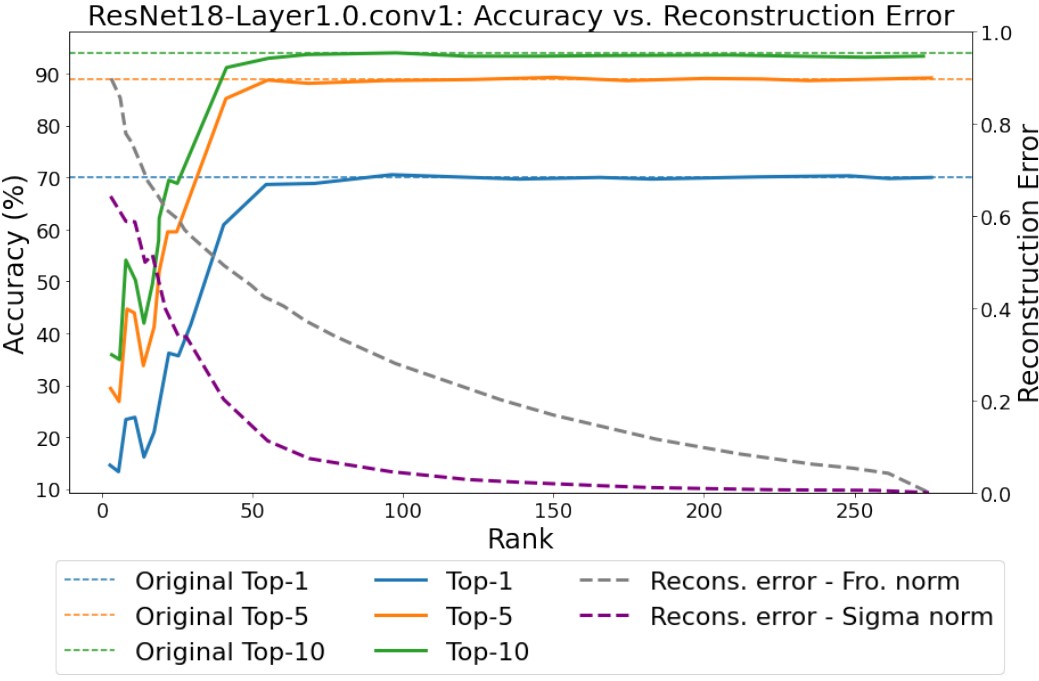

Figure 7: Relative reconstruction error and accuracy with respect to rank of the CP-decomposition of the second convolution of ResNet18.

In this section we explore the link between reconstruction error and accuracy. To do this we took a neural network trained on ImageNet and we decomposed one layer with the CP decomposition at different ranks. For each compression rank we computed the relative reconstruction error in Frobenius norm and distribution-aware norm. We also computed the accuracy of the model with the decomposed layer. The results are shown in Figure 7. We see that the accuracy of the model is not changing linearly with the rank, instead it first increases rapidly and then saturates. The Frobenius reconstruction error is more smooth and does not show a clear drop in reconstruction error when the accuracy increases. The distribution-aware reconstruction error is a bit chaotic for low-ranks. This is due to the fact that the optimisation is not done for this norm but for the Frobenius norm, hence we don't have the guarantee that the distribution-aware reconstruction error will decrease when the

rank increases. We observe that the distribution-aware reconstruction error is inversely correlated with the accuracy of the model. Indeed, for the low ranks, the distribution-aware reconstruction error rapidly decreases until the accuracy saturates. After that, the distribution-aware reconstruction error is still decreasing slowly while the accuracy is not changing. This shows that the distribution-aware reconstruction error is a good metric to measure the quality of the decomposition and that we could use it to choose the rank of the decomposition without having to compute the accuracy of the model.

The behavior observed in Figure 7 is representative of what we consistently observed for the first layers across all the networks tested. In addition, for the last layers we still have similar structure for the accuracy and the reconstruction error in sigma-norm but the Frobenius norm tends to be way more aligned with the sigma-norm.

