# OpenReview forum: "Distribution-Aware Tensor Decomposition for Compression of Convolutional Neural Networks"
_NeurIPS.cc/2025/Conference — NeurIPS 2025 poster_

### Official Review · Reviewer_2AV8 · 2025-06-16

**Clarity:** 3
**Significance:** 2
**Originality:** 3
**Rating:** 3
**Confidence:** 5

**Summary:**

This paper proposes a data-informed norm to measure the compression error in the function space, which leads to a lower accuracy drop compared to the traditional Frobenius norm. To optimize the new norm, the authors develop new alternating least square algorithms for Tucker-2 and CPD. Experiments show that this approach can outperform the Frobenius norm in terms of accuracy while eliminating the need of post-compression fine-tuning. Additionally, the method maintains its effectiveness even when optimized with unseen datasets, offering advantages for scenarios requiring data privacy protection.

**Questions:**

see weaknesses

**Ethical Concerns:**

["NO or VERY MINOR ethics concerns only"]

**Limitations:**

The authors clearly describe the limitations.

**Quality:**

3

**Strengths And Weaknesses:**

Strengths:
1. The paper is well-written and easy to follow.
2. The authors propose a new functional norm, which is straightforward to implement.
3. The authors provide a clear and detailed derivation of the optimization process under the new norm, and the comparative analysis between the ALS-based optimization algorithm and the tensor deflation strategy is well-justified.

Weaknesses:
1. The insight behind the proposed norm should be clarified. Since the difference between the proposed norm and the Frobenius norm appears to be the multiplication by the covariance matrix, the underlying reason why this modification improves performance should be explained.
2. The comparative experiments could be more comprehensive. In addition to comparing with Tucker2 and CP using the Frobenius norm in the weight space, it would be valuable to include comparisons with other data-driven compression methods. A basic additional comparison could be Tucker2 and CP with the Frobenius norm in the layer's output space.
3. According to Equation 3, the functional norm measures the layer's output approximation error. How does this differ from using the Frobenius norm in the layer's output space, which is a common approach in this field?
4. A minor point: On page 7, 'where p is the netork' contains a typo ('netork' should be 'network').

---

> ### Author Rebuttal · Authors · 2025-07-31
>
> Thank you for your comments. We believe there may have been some misunderstanding regarding our core contribution, and we’d like to take this opportunity to clarify the key points of our work.
>
> Our central technical contribution is the formal connection between the functional approximation norm and the Frobenius norm in the layer's output space. While using the output-space Frobenius norm as a proxy for functional accuracy might seem intuitive, it is often used heuristically in the literature without justification or scalability. To our knowledge, no prior work has explicitly established the equivalence we prove, and this result is what allows us to adapt efficient, widely-used algorithms such as Tucker2-ALS and CP-ALS to a data-aware objective.
>
> This is not a simple reinterpretation. Previous attempts to incorporate output-space norms into low-rank compression have either remained theoretical or faced significant scalability issues. The closest related work [1] we are aware of, which does explore a related goal, is restricted to small datasets (CIFAR10) and concludes that “...the results are encouraging but limited due to the computational bottleneck of solving large scale SDPs.” It does not address convolutional layers or provide a tractable implementation pathway for larger models. We will add this reference to our related work and discuss this in more details.
>
> In contrast, our contribution enables the practical use of data-aware compression through low rang factorization in real architectures, without requiring new, unproven solvers. We see this as a significant step forward: by identifying a theoretical equivalence and exploiting it computationally, we go from a conceptually appealing but impractical direction to a usable, scalable method with measurable performance gains.
>
> 1. About Clarifying the Insight Behind the Proposed Norm:
> The key insight behind this modification is that the covariance matrix captures the dependencies and variations in the data. By incorporating this matrix into the norm, we effectively "weight" the approximation by the importance of different features, allowing the decomposition to focus on the most relevant aspects of the data. In other words, the covariance matrix provides a data-aware bias, ensuring that the model captures the structures that matter most for the task at hand.
>
> 2. We don't have direct methods for comparison, as previous approaches were specialized to pruning and quantization, which are complementary to our method rather than competing with it. Additionally, some methods, like the one in [1], are not scalable and are limited to smaller datasets.
>
> 3. Please see the explanation above.
>
> 4. Thank you for pointing this out, we will correct it in the revised version.

---

> > ### Author Response · Authors · 2025-08-01
> >
> > We apologize for the oversight in our original rebuttal. The missing reference is:
> >
> > [1] Papadimitriou, D., & Jain, S. (2021, September). Data-driven low-rank neural network compression. In 2021 IEEE International Conference on Image Processing (ICIP) (pp. 3547–3551). IEEE.
> >
> > Thank you for your understanding.

---

> ### Author Response · Authors · 2025-08-07
> **Regarding Your Review: New Results on Data-Aware Comparisons**
>
> Dear Reviewer,
>
> Thank you again for your insightful review. We are writing to share new experimental results that provide a powerful, empirical answer to your questions.
>
> Concurrently with our initial rebuttal, another reviewer's feedback led to a direct comparison against other data-aware indicators. We have completed these experiments and present the methodology and results below, followed by an analysis connecting them to your original inquiry.
>
>
> **Methodology for New Comparative Experiments**
>
> For this comparative study, we focus on the Tucker decomposition, as its structure allows for a straightforward and rigorous algorithmic formulation of the weighted baseline methods. This provides the cleanest and most direct testbed for this specific analysis.
>
> To ensure a fair and rigorous comparison, we designed and implemented two strong, data-aware baseline methods based on the indicators suggested. Each baseline guides the Tucker decomposition by solving a specific Weighted Alternating Least Squares (WALS) problem, ensuring a rigorous, head-to-head comparison with our own approach.
>
> *   **Fisher-Weighted Low-Rank Approximation (FW-LRA):** This method solves the weighted decomposition $\min ||\mathbf{H}\_{\text{fisher}} \odot (\mathbf{W} - \mathbf{W}\_{\text{recon}})||\_F^2$, where $ \odot $ is the element-wise multiplication. The weighting tensor $\mathbf{H}_{\text{fisher}}$ is derived from the diagonal of the Fisher Information Matrix, adapting a well-known pruning metric to our decomposition task by prioritizing the preservation of weights most sensitive to the network's loss.
>
> *   **Activation-Guided Low-Rank Approximation (AG-LRA):** This method solves $\min ||\mathbf{H}\_{\text{act}} \odot (\mathbf{W} - \mathbf{W}\_{\text{recon}})||\_F^2$ where $\mathbf{H}_{\text{act}}$ is based on the mean absolute activation of each output channel, prioritizing filters that are most active during inference.
>
> For all data-aware methods, the weighting tensors were computed using a representative subset of the ImageNet training data. The baselines use a simple element-wise product to apply their heuristic weights. In contrast, our method optimizes a functional norm involving a more complex tensorial product. This required a non-trivial algorithmic derivation to create a tractable solver that directly models the structural interaction between the weights and the data covariance $\mathbf{\Sigma}$.
>
> **Comparative Experimental Results (Top-1 Accuracy on ImageNet)**
>
> The experiments were conducted without post-compression fine-tuning to isolate the raw quality of the methods.
>
>
> **Table 1: Performance on GoogLeNet**
> \\begin{array}{|c|c|c|c|c|c|}
> \\hline \\textbf{VBMF} & \\textbf{Compr. Ratio} & \\textbf{Tucker2 (Baseline)} & \\textbf{AG-LRA} & \\textbf{FW-LRA} & \\textbf{Our Method} \\\\
> \\hline
> 0.5 & 1.24 & 60.2\\% & 60.5\\% & 53.0\\% & \\textbf{66.3\\%} \\\\
> 0.6 & 1.33 & 52.5\\% & 54.9\\% & 35.8\\% & \\textbf{64.2\\%} \\\\
> 0.7 & 1.42 & 39.9\\% & 43.0\\% & 6.0\\%  & \\textbf{60.1\\%} \\\\
> 0.8 & 1.51 & 23.6\\% & 25.1\\% & 1.1\\%  & \\textbf{52.2\\%} \\\\
> \\hline
> \\end{array}
>
>
>
> **Table 2: Performance on ResNet-18**
> \\begin{array}{|c|c|c|c|c|c|}
> \\hline \\textbf{VBMF} & \\textbf{Compr. Ratio} & \\textbf{Tucker2 (Baseline)} & \\textbf{AG-LRA} & \\textbf{FW-LRA} & \\textbf{Our Method} \\\\
> \\hline
> 0.40 & 1.40 & 58.9\\% & 54.9\\% & 36.7\\% & \\textbf{66.8\\%} \\\\
> 0.45 & 1.50 & 52.2\\% & 45.0\\% & 18.9\\% & \\textbf{66.1\\%} \\\\
> 0.50 & 1.63 & 36.1\\% & 37.5\\% & 12.2\\% & \\textbf{64.9\\%} \\\\
> 0.55 & 1.77 & 30.2\\% & 31.2\\% & 4.2\\%  & \\textbf{63.3\\%} \\\\
> \\hline
> \\end{array}
>
>
> **Analysis: How These Results Address Your Initial Questions**
>
> As the data in the tables demonstrates, these new empirical results provide direct, quantitative answers to the insightful points you raised in your initial review. Specifically:
>
> 1. **The Insight Behind Our Norm:** The clear failure of the adapted heuristics provides a stark demonstration of why our covariance-based approach is necessary and not just an arbitrary modification. The data shows that simply being "data-aware" is not enough; the mechanism must be structurally sound.
>
> 2. **The Need for More Comprehensive Comparisons:** These tables provide the direct, head-to-head comparison that was needed, validating our method against strong, adapted baselines.
>
> 3. **The Difference from Other Objectives:** The results show a vast performance gap between our functionally-grounded norm and the adapted metrics, underscoring that the choice of objective is critically important.
>
> This new evidence perfectly complements our earlier theoretical clarification. Your review was instrumental in highlighting this comparative context, and we hope this data provides a thorough response to your concerns and strengthens our paper.
>
> Thank you again for your valuable engagement.
>
>
>
> Sincerely,
>
> The Authors

---

### Official Review · Reviewer_GT7A · 2025-06-17

**Clarity:** 3
**Significance:** 2
**Originality:** 2
**Rating:** 4
**Confidence:** 3

**Summary:**

This paper proposed a tensor decomposition method based on data-aware norms that does not require access to the original training data, aiming to improve the compression efficiency and practical applicability of convolutional neural networks in resource-constrained environments.

**Questions:**

1. The method proposed in the paper, which utilizes the distribution-aware norm based on the activation covariance matrix and its tensor decomposition, has achieved remarkable results, demonstrating the advantages of the data-driven approach. Are there any other forms of data-aware indicators to guide the compression process?

2. The text states that when the Sigma matrix is derived from a subset of the ImageNet training set, the performance of the Tucker2-ALS-Sigma algorithm is lower compared to when it is calculated on CIFAR-10 or CIFAR-100. The authors speculate that this might be related to the difference in image resolution. Based on this, apart from image resolution, are there any other factors that might affect the performance across different datasets? For example, differences in the category distribution of the datasets, the number of samples, or the diversity of image content?

3. Please read weakness 1

4. Please read weakness 2

**Ethical Concerns:**

["NO or VERY MINOR ethics concerns only"]

**Final Justification:**

For questions:

While the authors acknowledge the existence of other data-aware indicators and provide theoretical justification for their approach, a quantitative experimental comparison is still necessary. Even though methods like gradient-based importance (Fisher Information) and activation sparsity may not be directly designed for tensor decomposition, they can potentially be adapted or serve as baseline comparisons to demonstrate the superiority of the proposed method.

Could the authors please provide additional experimental results in a new comment that compare the proposed distribution-aware norm against these adapted baseline methods in terms of both accuracy and compression ratio?

The author provided the supplementary experimental results. These detailed experimental results and analyses have addressed my concerns.

**Limitations:**

yes

**Paper Formatting Concerns:**

In Table 1 and Table 2, the table numbers and titles do not appear before the tables.

**Quality:**

3

**Strengths And Weaknesses:**

Strength：
1. Introduces a covariance matrix derived from data distribution as a weighting norm, thereby converting traditional error measurements in weight space into function space metrics that better reflect perceptual error.

2. Achieves high accuracy without requiring fine-tuning on the original training dataset.

3. Maintains favorable performance even when applied across different datasets.

Weakness:
1. As shown in Figure 1, the compressed model using this method outperforms even the fine-tuned model in terms of accuracy, despite not undergoing any fine-tuning itself. This result is somewhat counterintuitive, as fine-tuning is generally expected to improve model performance by better adapting it to the data distribution. The paper provides limited explanation for this phenomenon.

2. The paper notes that when the covariance matrix Σ is computed from a subset of the ImageNet training set, the algorithm’s performance is slightly inferior to that obtained when Σ is calculated using the CIFAR-10 or CIFAR-100 datasets. The authors hypothesize that this discrepancy may be attributed to differences in image resolution. Nevertheless, this assertion is supported only by preliminary validation experiments and lacks comprehensive analysis and robust experimental evidence.

---

> ### Author Rebuttal · Authors · 2025-07-31
>
> 1.
>
> Thank you for your insightful question. There are indeed several data-aware indicators that have been explored in other compression settings, such as pruning and quantization. These indicators include gradient-based importance (e.g., Fisher Information) [1], activation sparsity [2], mutual information with outputs, and per-sample sensitivity [3]. While these techniques are valuable for guiding compression, they are typically focused on element-wise operations such as component removal or bit reduction, which are not directly applicable to tensor decomposition.
>
> Our method introduces a novel, functionally grounded indicator tailored specifically for low-rank factorization, shifting the focus from traditional reconstruction error to preserving task-relevant structure based on how the data actually flows through the model.
>
> That said, we recognize the broader landscape of data-aware compression methods, and we will mention techniques like pruning and quantization in the related work section. While these methods are not directly applicable to tensor decomposition, they provide useful context for how data-driven indicators are used in other compression strategies. This will help to position our method within the broader research field, and we see potential for exploring how insights from these techniques could inform future combinations or extensions of our approach.
>
> [1] Tu, M., Berisha, V., Cao, Y., & Seo, J. S.. Reducing the model order of deep neural networks using information theory. In 2016 I (ISVLSI) (pp. 93-98). IEEE.
>
> [2] Rhu, M., O'Connor, M., Chatterjee, N., Pool, J., Kwon, Y., & Keckler, S. W. . Compressing DMA engine: Leveraging activation sparsity for training deep neural networks. In 2018  (HPCA) . IEEE.
>
> [3] Chen, J., Zheng, L., Yao, Z., Wang, D., Stoica, I., Mahoney, M., & Gonzalez, J.. Actnn: Reducing training memory footprint via 2-bit activation compressed training. 2021 In ICML. PMLR.
>
> 2.
> Thank you for your insightful question. This is indeed a very relevant and practical inquiry, especially for users looking to apply our method in real-world settings where dataset choice for Sigma matrix computation can impact performance. While this aspect isn't the central focus of our work, we do recognize the value of providing practical guidelines to help users make informed decisions about which datasets to use.
>
> In response to your question, we conducted several supplementary experiments to explore how factors like dataset size, category distribution, and image diversity affect performance. These experiments are intended to offer valuable insights into the real-world applicability of our approach, though they are not central to the core algorithmic development and theoretical contributions of the paper.
>
> To investigate the effect of category distribution and image content diversity on our method, we experimented with subsets of ImageNet consisting of similar categories (e.g., Aircrafts, Dogs, or Reptiles) as well as mixes of images drawn from more diverse categories. We observed that using images from similar categories resulted in slightly reduced performance (at most -2%) compared to broader (mixed) category diversity.
>
> We also computed the Sigma matrix using datasets with limited category diversity, such as Oxford Flowers 102 and FGVC-Aircraft. Despite their smaller diversity, the results were comparable to those obtained using ImageNet, demonstrating that category distribution and image content diversity have only a minor impact on performance. Importantly, our approach consistently outperforms baseline methods that do not take data distribution into account.
>
> Finally, we evaluated the impact of the number of samples (5,000, 10,000, 20,000) used to compute the Sigma matrices by testing with varying sample sizes from the ImageNet training dataset. The performance of the Sigma algorithms was not significantly affected by the number of samples chosen (differences of at most +-0.1%), supporting the robustness of our method across different dataset sizes.
>
> We will include these additional results in the supplementary material, as we do not have enough space to present them in the answer.
>
> 3.
>
> Thank you for pointing this out. The result may indeed appear counter-intuitive at first glance, and we realize that the paper may not have been sufficiently explicit about the setup. To clarify: the fine-tuning was applied to the baseline (standard) compression method—not to our method. Despite this, our approach sometimes achieves better accuracy without any fine-tuning, which we agree warrants further explanation.
>
> The underlying reason lies in how each method approaches compression. Standard low-rank factorization treats all parts of the tensor equally, aiming to preserve global structure, often at the expense of compressing away information that is critical for generalization. Fine-tuning can partially recover performance, but only within the local basin of the degraded solution, it cannot bring back lost structure.
>
> In contrast, our method explicitly prioritizes the preservation of tensor components that are most impactful for the actual data distribution. By focusing the compression on the information that directly influences the model’s output on real inputs, it preserves more task-relevant capacity, even with the same tensor size. This likely explains why it can outperform the standard method, even post-fine-tuning.
>
> We are not claiming that our method always outperforms fine-tuned baselines (for example this is not the case in Figure 2), but we do believe this result highlights the strength of data-aware compression strategies, and we will add a clearer discussion of this effect in the revised manuscript.
>
> 4.
>
> Thank you for your insightful comment. We agree that the initial hypothesis regarding the effect of image resolution on performance was based on preliminary experiments. To address your concerns, we conducted a more comprehensive set of experiments to explore this further.
>
> Impact of Resolution on Performance:
> To investigate the potential role of resolution in the discrepancy between ImageNet and CIFAR-10 performance, for models trained on CIFAR-10, we conducted experiments where we reduced the resolution of both ImageNet and Aircraft images using bilinear downsampling. This simulates the effect of compression (e.g., JPEG) commonly seen with CIFAR-10 images. As shown in our supplementary results, we observed a significant improvement in performance on both ResNet18 and GoogLeNet, as we had intuitively expected. Upon analyzing the covariance matrices of the output layers, we found that the entropy of the eigenvalues was much closer to that observed when using CIFAR-10. Additionally, we computed the distance between the Sigma matrices for the different datasets and found them to be much more similar after downsampling. This suggests that the resolution difference does indeed impact the features extracted by the neural network, particularly in the first layers where low-level features such as edges are extracted.
>
> Upsampling Experiments:
> To further investigate the effect of resolution on models trained on ImageNet and tested on CIFAR-10 or CIFAR-100, we experimented with different upsampling methods. Specifically, we compared bilinear interpolation with bicubic interpolation for rescaling the images to match the ImageNet resolution. We observed that the use of bicubic interpolation resulted in better performance than bilinear interpolation, reinforcing our hypothesis that features extracted in the first layers are particularly sensitive to resolution differences. Despite this improvement, however, the performance was still lower compared to using images of similar resolution. This strongly suggests that resolution similarity is a critical factor, as it directly influences the representation learned by the model in the initial layers.
>
> These experiments, we believe, provide more robust evidence that resolution differences, particularly in the first layers of the network, have a significant impact on the performance of the proposed method. The results indicate that aligning resolution between the datasets is likely the most important factor when computing the covariance matrix, as it directly affects the features extracted by the network.
>
> **Table:** Accuracy comparison of the Tucker2-ALS-Sigma algorithm using Sigma matrices computed from different datasets on a ResNet-18 model trained with CIFAR-10. The Sigma matrices were derived from the full CIFAR-10 and CIFAR-100 training sets, as well as 50,000 ImageNet training images. For the ImageNet (Down) and  Aircraft (Down) variants, images were first downsampled to CIFAR-10 resolution ($3 \times 32 \times 32$) and then upsampled to ImageNet resolution ($3 \times 224 \times 224$) before computing the Sigma matrix, whereas for the other variants, images were only upsampled to ImageNet resolution.
>
> $$
> \\begin{array}{|c|c|c|c|c|c|c|c|}
> \\hline
> \\textbf{Ratio} & \\textbf{Compr Rate} & \\textbf{Sigma of Cifar 10 (Up)} & \\textbf{Sigma of Cifar 100 (Up)} & \\textbf{Sigma of ImageNet (Orig.)} & \\textbf{Sigma of ImageNet (Down)} & \\textbf{Sigma of Aircraft (Orig.)} & \\textbf{Sigma of Aircraft (Down)} \\\\
> \\hline
> 0.5 & 1.6 & 93.9 & 93.9 & 93.7 & 94.0 & 93.1 & 93.6 \\\\
> 0.6 & 1.89 & 93.7 & 93.7 & 93.4 & 93.7 & 92.1 & 93.3 \\\\
> 0.7 & 2.25 & 93.5 & 93.5 & 92.7 & 93.2 & 89.8 & 92.4 \\\\
> 0.8 & 2.73 & 93.0 & 92.8 & 91.0 & 92.5 & 86.9 & 91.1 \\\\
> 0.9 & 3.37 & 91.9 & 91.7 & 88.6 & 91.0 & 81.7 & 88.6 \\\\
> 1.0 & 4.22 & 90.2 & 89.5 & 83.5 & 87.9 & 72.0 & 82.7 \\\\
> \\hline
> \\end{array}
> $$
>
> 5.
>
> Thank you for pointing this out. We will correct the table numbering and ensure that titles appear properly before Tables 1 and 2 in the revised version.
>
>
> Please note that due to space limitations, we were unable to include all the new results in this response, but we will be adding them to the paper. Feel free to ask us to put them here in the second phase.

---

> > ### Comment · Reviewer_GT7A · 2025-08-05
> > **Official Comment by Reviewer GT7A**
> >
> > Thank you for your reply. I still have a few concerns regarding your responses to Reply 1 and Reply 2:
> >
> > 1. While the authors acknowledge the existence of other data-aware indicators and provide theoretical justification for their approach, a quantitative experimental comparison is still necessary. Even though methods like gradient-based importance (Fisher Information) and activation sparsity may not be directly designed for tensor decomposition, they can potentially be adapted or serve as baseline comparisons to demonstrate the superiority of the proposed method.
> >
> > 2. Could the authors please provide additional experimental results in a new comment that compare the proposed distribution-aware norm against these adapted baseline methods in terms of both accuracy and compression ratio?
> >
> > Based on the above concerns, I will maintain the current score temporarily. If the authors can adequately address these issues with comparative experimental evidence, I will consider increasing the score accordingly.

---

> > > ### Author Response · Authors · 2025-08-07
> > >
> > > Dear Reviewer,
> > >
> > > Thank you for the follow-up and the very interesting suggestion. We agree that a quantitative comparison against other data-aware indicators is essential for contextualizing our work. We have performed the requested experiments, which provided significant new insights.
> > >
> > > **Methodology for Adapted Baselines**
> > >
> > > For this comparative study, we focus on the Tucker decomposition, as its structure allows for a straightforward and rigorous algorithmic formulation of the weighted baseline methods. This provides the cleanest and most direct testbed for this specific analysis.
> > >
> > > We designed and implemented two strong, data-aware baseline methods based on the indicators you suggested. Each baseline guides the Tucker decomposition by solving a specific Weighted Alternating Least Squares (WALS) problem, ensuring a rigorous, head-to-head comparison with our own approach.
> > >
> > > *   **Fisher-Weighted Low-Rank Approximation (FW-LRA):** Solves the weighted decomposition $\min ||\mathbf{H}\_{\text{fisher}} \odot (\mathbf{W} - \mathbf{W}\_{\text{recon}})||\_F^2$, where $ \odot $ is the element-wise multiplication. The weighting tensor $\mathbf{H}_{\text{fisher}}$ is derived from the diagonal of the Fisher Information Matrix, adapting a well-known pruning metric to our decomposition task by prioritizing the preservation of weights most sensitive to the network's loss.
> > >
> > > *   **Activation-Guided Low-Rank Approximation (AG-LRA):** Solves $\min ||\mathbf{H}\_{\text{act}} \odot (\mathbf{W} - \mathbf{W}\_{\text{recon}})||\_F^2$ where $\mathbf{H}_{\text{act}}$ is based on the mean absolute activation of each output channel, prioritizing filters that are most active during inference.
> > >
> > > For all data-aware methods, the weighting tensors were computed using a representative subset of the ImageNet training data. The baselines use a simple element-wise product to apply their heuristic weights.  In contrast, our method optimizes a functional norm involving a more complex tensorial product. This required a non-trivial algorithmic derivation to create a tractable solver that directly models the structural interaction between the weights and the data covariance $\mathbf{\Sigma}$.
> > >
> > >
> > > **Comparative Experimental Results (Top-1 Accuracy on ImageNet)**
> > >
> > > The experiments were conducted without post-compression fine-tuning to isolate the raw quality of the methods.
> > >
> > > **Table 1: Performance on GoogLeNet**
> > > \\begin{array}{|c|c|c|c|c|c|}
> > > \\hline \\textbf{VBMF} & \\textbf{Compr. Ratio} & \\textbf{Tucker2 (Baseline)} & \\textbf{AG-LRA} & \\textbf{FW-LRA} & \\textbf{Our Method} \\\\
> > > \\hline
> > > 0.5 & 1.24 & 60.2\\% & 60.5\\% & 53.0\\% & \\textbf{66.3\\%} \\\\
> > > 0.6 & 1.33 & 52.5\\% & 54.9\\% & 35.8\\% & \\textbf{64.2\\%} \\\\
> > > 0.7 & 1.42 & 39.9\\% & 43.0\\% & 6.0\\%  & \\textbf{60.1\\%} \\\\
> > > 0.8 & 1.51 & 23.6\\% & 25.1\\% & 1.1\\%  & \\textbf{52.2\\%} \\\\
> > > \\hline
> > > \\end{array}
> > >
> > >
> > >
> > > **Table 2: Performance on ResNet-18**
> > > \\begin{array}{|c|c|c|c|c|c|}
> > > \\hline \\textbf{VBMF} & \\textbf{Compr. Ratio} & \\textbf{Tucker2 (Baseline)} & \\textbf{AG-LRA} & \\textbf{FW-LRA} & \\textbf{Our Method} \\\\
> > > \\hline
> > > 0.40 & 1.40 & 58.9\\% & 54.9\\% & 36.7\\% & \\textbf{66.8\\%} \\\\
> > > 0.45 & 1.50 & 52.2\\% & 45.0\\% & 18.9\\% & \\textbf{66.1\\%} \\\\
> > > 0.50 & 1.63 & 36.1\\% & 37.5\\% & 12.2\\% & \\textbf{64.9\\%} \\\\
> > > 0.55 & 1.77 & 30.2\\% & 31.2\\% & 4.2\\%  & \\textbf{63.3\\%} \\\\
> > > \\hline
> > > \\end{array}
> > >
> > >
> > >
> > > **Analysis of Results**
> > >
> > > The empirical results are unambiguous. Our method consistently and substantially outperforms all baselines. More importantly, the performance of the adapted baselines reveals a crucial insight:
> > >
> > > The direct application of data-aware heuristics from other domains, such as pruning, is not a reliable strategy for tensor decomposition. The FW-LRA method, based on a globally-scoped metric, leads to a severe degradation in performance. The AG-LRA method exhibits inconsistent behavior, sometimes offering marginal gains and other times performing worse than the data-agnostic baseline, suggesting it lacks robustness.
> > >
> > >
> > > In stark contrast, our method **consistently and substantially outperforms all baselines** across every tested model and compression setting. These results demonstrate the significant value of our paper's core contribution: a principled, data-aware norm and the tractable algorithm developed to optimize it. The comparison shows this integrated design achieves a level of robustness and performance not matched when adapting general-purpose heuristics from other compression domains.
> > >
> > > **Plan for Revision**
> > >
> > > We will incorporate this powerful and nuanced comparative analysis, including the methodology for the adapted baselines, into the Experiments section of our paper.
> > >
> > > Thank you again for your constructive feedback, which has led to a significant strengthening of our paper's experimental validation. We believe this comprehensive data provides a thorough response to your final concern and hope it will merit a positive re-evaluation.
> > >
> > > Sincerely,
> > >
> > > The Authors

---

> ### Author Response · Authors · 2025-08-07
> **Supplementary experimental data Part 1**
>
> Dear Reviewer,
>
> As promised in our initial rebuttal, we are providing the supplementary experimental data to fully address your insightful questions. Due to space constraints, we could not include these tables in our first response. We believe this comprehensive data provides the robust evidence requested for our claims.
>
> ---
>
> ### **1. Rigorously Testing the Resolution Hypothesis (addressing Weakness 2)**
>
> Our initial rebuttal proposed a hypothesis: that differences in image resolution were a primary factor in cross-dataset performance. The following experiments were designed to test this rigorously, directly addressing the concern that our initial assertion lacked comprehensive validation.
>
> *   **Key Finding 1 (Performance Recovery):** The data in Tables 1 and 2 demonstrates a significant performance recovery when the resolution of proxy datasets (e.g., ImageNet) is matched to the target dataset's resolution (CIFAR-10) via downsampling. For example, in Table 1, **Sigma of ImageNet (Down)** consistently outperforms **Sigma of ImageNet (Orig.)**.
> *   **Key Finding 2 (Upsampling Quality):** Table 3 further shows that higher-quality upsampling methods (Bicubic vs. Bilinear) yield better performance, reinforcing that our method is sensitive to the feature quality preserved during scaling.
>
> These results provide strong empirical support for our claim that resolution alignment is a critical factor for optimal performance.
>
>
> **Table 1** Accuracy comparison on **ResNet-18 (trained on CIFAR-10)** using Sigma matrices computed from various datasets and resolution-matching strategies.
> $$
> \\begin{array}{|c|c|c|c|c|c|}
> \\hline \\textbf{VBMF Ratio} & \\textbf{Compr. Rate} & \\textbf{$\\Sigma$ of CIFAR-10} & \\textbf{$\\Sigma$ of CIFAR-100} & \\textbf{$\\Sigma$ of ImageNet (Orig.)} & \\textbf{$\\Sigma$ of ImageNet (Down)} & \\textbf{Sigma of Aircraft (Orig.)} & \\textbf{Sigma of Aircraft (Down)} \\\\
> \\hline
> 0.5 & 1.6 & 93.9 & 93.9 & 93.7 & 94.0 & 93.1 & 93.6 \\\\
> \\hline
> 0.6 & 1.89 & 93.7 & 93.7 & 93.4 & 93.7 & 92.1 & 93.3 \\\\
> \\hline
> 0.7 & 2.25 & 93.5 & 93.5 & 92.7 & 93.2 & 89.8 & 92.4 \\\\
> \\hline
> 0.8 & 2.73 & 93.0 & 92.8 & 91.0 & 92.5 & 86.9 & 91.1 \\\\
> \\hline
> 0.9 & 3.37 & 91.9 & 91.7 & 88.6 & 91.0 & 81.7 & 88.6 \\\\
> \\hline
> 1.0 & 4.22 & 90.2 & 89.5 & 83.5 & 87.9 & 72.0 & 82.7 \\\\
> \\hline
> \\end{array}
> $$
>
>
> **Table 2** Accuracy comparison on **GoogLeNet (trained on CIFAR-10)** using Sigma matrices computed from various datasets and resolution-matching strategies.
> $$
> \\begin{array}{|c|c|c|c|c|c|}
> \\hline \\textbf{VBMF Ratio} & \\textbf{Compr. Rate} & \\textbf{$\\Sigma$ of CIFAR-10} & \\textbf{$\\Sigma$ of CIFAR-100} & \\textbf{$\\Sigma$ of ImageNet (Orig.)} & \\textbf{$\\Sigma$ of ImageNet (Down)} \\\\
> \\hline
> 0.6 & 1.39 & 94.0 & 93.8 & 92.5 & 93.4 \\\\
> \\hline
> 0.7 & 1.51 & 93.4 & 93.1 & 90.7 & 92.3 \\\\
> \\hline
> 0.8 & 1.64 & 92.6 & 92.0 & 88.1 & 90.4 \\\\
> \\hline
> 0.9 & 1.76 & 90.4 & 89.3 & 81.7 & 87.0 \\\\
> \\hline
> 0.95 & 1.83 & 88.5 & 87.1 & 76.9 & 83.3 \\\\
> \\hline
> 1.0 & 1.89 & 86.3 & 84.2 & 70 & 78.6 \\\\
> \\hline
> \\end{array}
> $$
>
>
>
>
>
>
> **Table 3** Accuracy comparison on **ResNet-18 (trained on ImageNet)**, showing the impact of upsampling quality when using low-resolution datasets for Sigma computation.
> $$
> \\begin{array}{|c|c|c|c|c|c|}
> \\hline
> \\textbf{VBMF Ratio} &
> \\textbf{Compr. Rate} &
> \\textbf{$\Sigma$ of CIFAR-10 (Bilinear)} &
> \\textbf{$\Sigma$ of CIFAR-10 (Bicubic)} &
> \\textbf{$\Sigma$ of CIFAR-100 (Bilinear)} &
> \\textbf{$\Sigma$ of CIFAR-100 (Bicubic)} \\\\
> \\hline
> 0.4 & 1.4 & 55.4 & 56.3 & 55.6 & 56.7 \\\\
> \\hline
> 0.45 & 1.5 & 52.3 & 53.8 & 52.4 & 53.9 \\\\
> \\hline
> 0.5 & 1.63 & 47.2 & 50.5 & 48.3 & 50.8 \\\\
> \\hline
> 0.55 & 1.77 & 41.7 & 46.5 & 42.0 & 47.0 \\\\
> \\hline
> \\end{array}
> $$

---

> > ### Author Response · Authors · 2025-08-07
> > **Supplementary experimental data Part 2**
> >
> > ### **2.  Investigating Other Potential Factors (addressing Question 2)**
> >
> >
> > To address your question about factors beyond resolution, we investigated the impact of dataset diversity, semantic coherence, and sample size.
> >
> > *   **Key Finding 1 (Broad Diversity):** Table 4 shows that while a large, diverse dataset like ImageNet is optimal for computing Sigma, performance remains strong even with smaller, more specialized datasets.
> > *   **Key Finding 2 (Semantic Coherence):** Table 5 reveals that using a dataset of thematically-grouped images (e.g., multiple dog breeds) versus a mix of diverse categories has only a minor impact on performance, highlighting the method's flexibility.
> > *   **Key Finding 3 (Sample Size):** Tables 6 and 7 show that our method is remarkably robust to the number of samples used for covariance estimation, demonstrating its practical stability.
> >
> > Taken together, these results indicate that our method is highly robust to variations in sample size and diversity, with resolution alignment being the most critical factor for cross-dataset transfer.
> >
> >
> >
> >
> >
> >
> >
> >
> > **Table 4** Impact of using specialized datasets (less diverse than ImageNet) for Sigma computation on **ResNet-18**.
> > $$
> > \\begin{array}{|c|c|c|c|c|c|}
> > \\hline
> > \\textbf{VBMF Ratio} & \\textbf{Compr. Rate} & \\textbf{$\\Sigma$ of ImageNet} & \\textbf{$\\Sigma$ of Aircraft} & \\textbf{$\\Sigma$ of Flowers-102} \\\\
> > \\hline
> > 0.4 & 1.4 & 66.3 & 64.2 & 64.6 \\\\
> > \\hline
> > 0.45 & 1.5 & 65.2 & 62.4 & 63.2 \\\\
> > \\hline
> > 0.5 & 1.63 & 63.9 & 59.9 & 61.6 \\\\
> > \\hline
> > 0.55 & 1.77 & 61.9 & 56.5 & 59.1 \\\\
> > \\hline
> > \\end{array}
> > $$
> >
> >
> >
> > **Table 5** Impact of semantic coherence in the Sigma computation dataset on **ResNet-18** accuracy.
> > $$
> > \\begin{array}{|c|c|c|c|c|c|}
> > \\hline
> > \\textbf{VBMF Ratio} & \\textbf{Compr. Rate}  & \\textbf{$\Sigma$ from 'Aircraft' Classes} & \\textbf{$\Sigma$ from 'Dog' Classes} & \\textbf{$\Sigma$ from 'Reptile' Classes} & \\textbf{$\Sigma$ from Diverse Classes} \\\\
> > \\hline
> > 0.4 & 1.4 & 65.6 & 66.1 & 65.3 & 66.6 \\\\
> > \\hline
> > 0.45 & 1.5 & 64.3 & 65 & 64.2 & 65.7 \\\\
> > \\hline
> > 0.5 & 1.63 & 62.6 & 63.7 & 62.3 & 64.4 \\\\
> > \\hline
> > 0.55 & 1.77 & 60 & 61.7 & 60.2 & 62.6 \\\\
> > \\hline
> > \\end{array}
> > $$
> >
> >
> >
> > **Table 6** Robustness to sample size for Sigma computation on **ResNet-18**. Results are mean ± stddev over 5 seeds.
> > \\begin{array}{|c|c|c|c|c|c|}
> > \\hline \\textbf{VBMF Ratio} & \\textbf{Compr. Rate} & \\textbf{\$\\Sigma\$ of ImageNet-5k} & \\textbf{\$\\Sigma\$ of ImageNet-10k} & \\textbf{\$\\Sigma\$ of ImageNet-20k} & \\textbf{\$\\Sigma\$ of ImageNet-50k} \\\\
> > \\hline
> > 0.40 & 1.40 & 66.81 \\pm 0.02 & 66.80 \\pm 0.03 & 66.80 \\pm 0.02 & 66.80 \\pm 0.02 \\\\
> > \\hline
> > 0.45 & 1.50 & 66.09 \\pm 0.03 & 66.11 \\pm 0.06 & 66.10 \\pm 0.02 & 66.08 \\pm 0.02 \\\\
> > \\hline
> > 0.50 & 1.63 & 64.88 \\pm 0.03 & 64.90 \\pm 0.05 & 64.91 \\pm 0.02 & 64.89 \\pm 0.01 \\\\
> > \\hline
> > 0.55 & 1.77 & 63.25 \\pm 0.05 & 63.29 \\pm 0.07 & 63.31 \\pm 0.03 & 63.30 \\pm 0.01 \\\\
> > \\hline
> > \\end{array}
> >
> >
> > **Table 7** Robustness to sample size for Sigma computation on **GoogLeNet**. Results are mean ± stddev over 5 seeds.
> > \\begin{array}{|c|c|c|c|c|c|}
> > \\hline \\textbf{VBMF Ratio} & \\textbf{Compr. Rate} & \\textbf{\$\\Sigma\$ of ImageNet-5k} & \\textbf{\$\\Sigma\$ of ImageNet-10k} & \\textbf{\$\\Sigma\$ of ImageNet-20k} & \\textbf{\$\\Sigma\$ of ImageNet-50k} \\\\
> > \\hline
> > 0.60 & 1.33 & 64.17 \\pm 0.04 & 64.13 \\pm 0.05 & 64.16 \\pm 0.07 & 64.17 \\pm 0.03 \\\\
> > \\hline
> > 0.70 & 1.42 & 60.04 \\pm 0.03 & 60.11 \\pm 0.06 & 60.13 \\pm 0.04 & 60.11 \\pm 0.04 \\\\
> > \\hline
> > 0.80 & 1.51 & 52.21 \\pm 0.09 & 52.18 \\pm 0.08 & 52.20 \\pm 0.08 & 52.17 \\pm 0.05 \\\\
> > \\hline
> > \\end{array}

---

> ### Comment · Reviewer_GT7A · 2025-08-07
> **Official Comment by Reviewer GT7A**
>
> Thank you for providing the supplementary experimental results. These detailed experimental results and analyses have resolved my doubts. I will raise my rating. I hope the author will include the subsequent experimental results in the manuscript.

---

> > ### Author Response · Authors · 2025-08-08
> >
> > Dear Reviewer,
> >
> > Thank you for your positive feedback and for confirming that our new results have addressed your concerns. We are grateful for your constructive engagement throughout the review process.
> >
> > We will incorporate these new experiments into the final manuscript. We agree that they significantly strengthen the paper’s empirical foundation, and we sincerely appreciate your suggestion that led to this addition.
> >
> > Sincerely,
> >
> > The Authors

---

### Official Review · Reviewer_qrJc · 2025-07-03

**Clarity:** 4
**Significance:** 3
**Originality:** 2
**Rating:** 5
**Confidence:** 3

**Summary:**

The paper introduced a novel tensor decomposition compression method for CNNs. The key contribution is to replace the Frobenius norm with the so-called Sigma norm, which is the root mean squared residual over the dataset of the compressed and uncompressed network. This puts emphasis on network weights that affect the output more. The author also showed that, for convolutional layers, this norm is also equivalent to the Frobenius norm rescaled with the data covariance. The author then derived the CP and Tucker-2 ALS under the Sigma norm. The author then demonstrated empirically that using the data-aware Sigma norm greatly improves the reconstruction accuracy.

**Questions:**

- Can your method be generalized beyond convolutional networks?
- How did you compute the covariance in equation 4, and what's the computational complexity?
- Your methods are kind of similar to compressed sensing. Could CS theory provide more theoretical guarantees or better covariance estimation strategies?

**Ethical Concerns:**

["NO or VERY MINOR ethics concerns only"]

**Final Justification:**

I will maintain my initial score for the paper. My initial concerns and questions are all well addressed, and the authors have conducted new experiments and obtained new findings as suggested by other reviewers.

**Limitations:**

Yes

**Quality:**

3

**Strengths And Weaknesses:**

Strengths:
- The method is mathematically well-founded, and the motivation of making the tensor compression data-aware makes a lot of sense
- The writing was very clear.
- Empirical performance was good. At the same rank, the proposed method is better than the baseline in terms of accuracy.

Weaknesses:
- The computational complexity of the method was not discussed. For example, the computation of the covariance matrix could scale badly, which could prevent this method from being applied to large networks.
- Some notations were not introduced, for example, the Khatri-Rao product
- Using data distribution as functional norm is not new

---

> ### Author Rebuttal · Authors · 2025-07-31
>
> 1. Can your method be generalized beyond convolutional networks?
>
> Thank you for raising this important point. Our method is designed to target convolutional layers, leveraging their unique structural properties. This method is not directly applicable to non-convolutional architectures like Transformers without some adaptation.
>
> The core idea of low-rank factorization itself is not constrained to CNNs. In fact, we’ve successfully applied our method to fully connected layers (e.g., in AlexNet to address a similar concern from the reviewer ZcfM), which have similar structural properties to those found in Transformer models, such as dense layers. This required only minor adjustments to our approach, suggesting that our method can be adapted to other linear layers with minimal changes.
>
> However, applying our method to Transformer architectures presents additional challenges due to the attention mechanisms and the non-local dependencies within the model, which do not exhibit the same kind of spatial locality that convolutions do. These structural differences require careful consideration in adapting our low-rank factorization technique to handle attention layers and other components of Transformers.
>
> While we have not yet explored these adaptations experimentally, we see this as a promising next step. One of the motivations for focusing on CNNs in our current work was to gain rigorous insight into the behavior of low-rank decompositions in practical, performance-critical scenarios. These insights will be critical in addressing the specific challenges posed by Transformer models in future work.
>
> We also note that CNNs remain widely used in real-world applications, especially in edge devices and memory-constrained environments, where this method already has high practical impact. Additionally, our approach is hardware-agnostic and can be stacked with other compression methods, including those currently applied to Transformer models.
>
> Finally, as Transformer models continue to grow in size and energy consumption, efficient compression remains a critical research priority—not just for performance, but also for sustainability. Our future work will build on this foundation, with a focus on bringing lightweight, interpretable compression to Transformer architectures in a similarly orthogonal and composable way.
>
> We’ll make this trajectory clearer in the paper and explicitly mention the applicability of our method to linear layers in Transformers, as well as our plans to extend this work in that direction.
>
> 2. How did you compute the covariance in equation 4, and what's the computational complexity?
>
> The cost (total number of operations) is quadratic in the number of features $(S \times H \times W)$ of a layer:
> for an intermediate representation of size $S \times H_i \times W_i$ and $N$ samples, the cost will be $N \times H_i \times W_i \times (S \times H \times W)^2$.
>
> This computational cost is efficiently parallelizable, which makes the process suitable for modern hardware, especially GPUs. The memory bandwidth required for these operations is  $(S \times H \times W)^2$, which is manageable given the memory architecture of contemporary systems.
>
> Importantly, this factorization step is performed **once per model** and **dataset**, meaning that it is a **one-time pre-processing step**. As a result, the cost of this operation is negligible compared to the long-term benefits of compression, which include reductions in inference time, computational cost, memory usage, and energy consumption during deployment
>
> 3. Your methods are kind of similar to compressed sensing. Could CS theory provide more theoretical guarantees or better covariance estimation strategies?
>
> Thank you for the insightful question. While our method shares similarities with compressed sensing (CS) in reducing redundancy, our approach focuses on exploiting low-rank structures in weight matrices, whereas CS primarily deals with sparse recovery of signals. The mathematical foundations are different, as CS relies on sparsity and can recover signals from fewer measurements, while we optimize the Frobenius norm in the output space to capture redundancies in neural networks.
>
> That said, compressed sensing theory could potentially provide insights into covariance estimation if the covariance structure of layer outputs is sparse. In such cases, CS could help reduce dimensionality and accelerate covariance computation by focusing on the most important components, using techniques like random projections or sparse recovery. We plan to explore whether CS-inspired techniques could help improve our method, especially in the context of combining our approach with pruning to enhance model efficiency.
>
> We plan to briefly mention compressed sensing in the related work section, specifically in relation to its use in sparse recovery and covariance estimation, while noting that it offers an interesting avenue for future research, particularly for sparsity-aware compression.

---

> > ### Comment · Reviewer_qrJc · 2025-08-08
> >
> > Thank you for your detailed rebuttal. My initial concerns and questions are all addressed. I have also read the discussion with other reviewers, which is also insightful.

---

> > > ### Author Response · Authors · 2025-08-08
> > >
> > > Dear Reviewer,
> > >
> > > Thank you for your feedback and for confirming that your concerns have been addressed. We are especially grateful that you took the time to read the full discussion and found it valuable. Your engagement has been instrumental in improving the paper.
> > >
> > > Sincerely,
> > >
> > > The Authors

---

### Official Review · Reviewer_ZcfM · 2025-07-07

**Clarity:** 4
**Significance:** 2
**Originality:** 3
**Rating:** 5
**Confidence:** 2

**Summary:**

This paper introduces distribution-aware tensor decomposition for neural network compression, replacing traditional weight-space optimization with function-space optimization using a data-informed norm that incorporates input covariance statistics. The proposed CP-ALS-Sigma and Tucker2-ALS-Sigma algorithms achieve competitive accuracy without fine-tuning and demonstrate transferability across datasets, representing a meaningful theoretical advancement in compression techniques that addresses practical deployment scenarios where original training data is unavailable.

**Questions:**

### Regarding Limited Experimental Scope:
1. Have you tested your method on modern architectures like MobileNet or EfficientNet that are commonly used in resource-constrained environments?

### Regarding Insufficient Baseline Comparisons:
2. How does your method compare against popular quantization techniques like INT8 or mixed-precision compression?

### Regarding Missing Statistical Rigor:
3. Could you provide error bars or confidence intervals by running experiments multiple times with different random seeds?

**Ethical Concerns:**

["NO or VERY MINOR ethics concerns only"]

**Limitations:**

yes

**Quality:**

4

**Strengths And Weaknesses:**

## Strengths

1. The paper introduces a principled shift from weight-space to function-space optimization by incorporating data distribution into tensor decomposition norms.

2. The proposed CP-ALS-Sigma and Tucker2-ALS-Sigma algorithms provide concrete implementations that directly optimize the distribution-aware norm with closed-form solutions.

3. The demonstrated ability to transfer covariance statistics across different datasets addresses real-world scenarios where original training data is unavailable.

## Weaknesses

1. Testing only three CNN architectures (ResNet-18/50, GoogLeNet) on primarily image classification tasks fails to demonstrate broad applicability of the method.

2. The paper lacks comparison with modern compression techniques like quantization, knowledge distillation, and other state-of-the-art methods beyond traditional tensor decomposition.

3. The absence of error bars, confidence intervals, or multiple-run statistics undermines the reliability and significance of the reported results.

---

> ### Author Rebuttal · Authors · 2025-07-31
>
> 1. Have you tested your method on modern architectures like MobileNet or EfficientNet that are commonly used in resource-constrained environments?
>
> We appreciate the reviewer’s suggestion regarding MobileNet and EfficientNet. These architectures are indeed optimized for efficiency at design time, using techniques such as depthwise separable convolutions and compound scaling. As a result, they are inherently low in redundancy, offering limited headroom for further compression via low-rank methods. Our method is designed for overparameterized models, where structural redundancy remains exploitable and compression yields more substantial gains.
>
> That said, we recognize the importance of evaluating generality. To this end, we have added results on AlexNet, which differs significantly from the ResNet and GoogLeNet families in both structure and parameter distribution. Notably, we applied our method to both convolutional and fully connected layers (which can be easily adapted from our original approach to convolutional layers), demonstrating its effectiveness across different types of layers. The results are summarized in Table and further support the applicability of our approach beyond the models initially reported.
>
> We also emphasize that in practice, there are two complementary paths to efficiency: (1) designing compact models from scratch, and (2) compressing high-performing models post hoc. Our work addresses the latter, a crucial use case in real-world deployments where retraining or architectural redesign is impractical. Continued advancement in post hoc compression techniques, like ours, enables broader applicability across legacy models, diverse architectures, and varying deployment constraints.
>
> **Table:** Top-1 accuracies of the compressed AlexNet architecture using Tucker2-ALS followed by SVD, and Tucker2-ALS-Sigma followed by SVD-Sigma, where the Sigma matrix is computed using 5,000 images from the ImageNet training dataset. AlexNet original Top-1 accuracy is 56%.
>
>
> $$
> \\begin{array}{|c|c|c|c|}
> \\hline
> \\textbf{VBMF Ratio} & \\textbf{Compr. Rate} & \\textbf{Tucker2 Sigma + SVD Sigma} & \\textbf{Tucker2 + SVD} \\\\
> \\hline
> 0.6 & 1.46 & 49.8 & 42.4 \\\\
> 0.65 & 1.63 & 48.1 & 40.1 \\\\
> 0.7 & 1.85 & 46.1 & 34.4 \\\\
> 0.8 & 2.54 & 40.9 & 25.8 \\\\
> \\hline
> \\end{array}
> $$
>
>
> 2. How does your method compare against popular quantization techniques like INT8 or mixed-precision compression?
>
> Quantization and low-rank factorization target different dimensions of model compression and are complementary rather than competing methods. Our work focuses on exploiting the linear structure in weight tensors through low-rank factorization, which reduces model size and computation without imposing hardware constraints or altering the underlying data distribution. This makes low-rank factorization an ideal candidate to be combined with quantization to further enhance model efficiency.
>
> In response to the reviewer's concern, we have added experiments where we quantized the compressed models obtained through low-rank factorization. As shown in Table, the performance of the compressed and quantized models remains preserved, demonstrating that the combination of these two methods provides a synergistic effect, enhancing efficiency without sacrificing accuracy.
>
> While quantization methods like INT8 reduce model size by focusing on numerical precision, low-rank factorization operates at a structural level, targeting redundancy in weight tensors. Thus, the two methods address orthogonal aspects of compression. Existing research and preliminary findings support the notion that combining these approaches can lead to even greater compression gains than applying either method alone [1].
>
> Though we have not yet explored integration with more advanced, distribution-aware quantization techniques, we believe this is a promising direction for future work. For now, we hope our findings provide a solid foundation for the complementary use of low-rank factorization and quantization, highlighting how these methods can work together to achieve broader deployment flexibility across different hardware and architectures.
>
> [1] Gui, S., Wang, H., Yang, H., Yu, C., Wang, Z., & Liu, J. (2019). Model compression with adversarial robustness: A unified optimization framework. Advances in neural information processing systems, 32.
>
> [2] Cherniuk, D., Abukhovich, S., Phan, A. H., Oseledets, I., Cichocki, A., & Gusak, J. (2024). Quantization aware factorization for deep neural network compression. Journal of Artificial Intelligence Research, 81, 973-988.
>
> **Table:** We present the accuracy results of the GoogleNet model compressed using Tucker2-ALS and Tucker2-ALS-Sigma algorithms, followed by post-compression quantization with FP16 and INT8 precisions including FP32 compressed models accuracies varying VBMF ratios.
> $$
> \\begin{array}{|c|c|c|c|c|c|c|}
> \\hline
> \\textbf{VBMF} & \\textbf{Tucker2 Sigma}&\\textbf{Tucker2 Sigma} &\\textbf{Tucker2 Sigma} & {\\textbf{Tucker2}}&{\\textbf{Tucker2}}& {\\textbf{Tucker2}}\\\\
> \\textbf{Ratio} & \\textbf{FP32} & \\textbf{FP16} & \\textbf{INT8} & \\textbf{FP32} & \\textbf{FP16} & \\textbf{INT8} \\\\
> \\hline
> 0.6  & 93.97 & 93.94 & 93.65 & 92.81 & 92.82 & 92.47 \\\\
> 0.7  & 93.43 & 93.39 & 93.19 & 91.16 & 91.18 & 91.24 \\\\
> 0.8  & 92.69 & 92.68 & 92.29 & 87.66 & 87.68 & 88.38 \\\\
> 0.9  & 90.65 & 90.68 & 90.02 & 76.43 & 76.34 & 77.03 \\\\
> 0.95 & 88.77 & 88.79 & 88.19 & 43.57 & 43.56 & 44.31 \\\\
> 1.0  & 86.73 & 86.73 & 85.91 & 33.73 & 33.71 & 34.33 \\\\
> \\hline
> \\end{array}
> $$
>
> 3. Could you provide error bars or confidence intervals by running experiments multiple times with different random seeds?
>
> Our Tucker2 experiments are deterministic, as we use a fixed initialization strategy (specifically, SVD of different matrices obtained from the weight tensor) and apply the generalized minimal residual method (GMRES), which is also deterministic. As a result, these experiments do not involve any random variability, and error bars or confidence intervals are not applicable here.
>
> However, for the CP method in the supplementary material, we used a random initialization, and the results do vary slightly across runs. To address this, we conducted 10 experiments with different random seeds using the GoogLeNet model trained on the CIFAR-10 dataset, applying the CP-ALS algorithm with varying VBMF ratios. For a VBMF ratio of 0.9, the mean Top-1 accuracy across 10 runs with different random seeds was 93.062\%, with a standard deviation of 0.145\% and a 95\% confidence interval of (92.958\%, 93.166\%). With a VBMF ratio of 0.95, the mean accuracy was 89.136\%, standard deviation 0.615\%, and confidence interval (88.696\%, 89.576\%). At a VBMF ratio of 1.0, the mean accuracy was 37.705\%, standard deviation 3.001\%, and confidence interval (35.558\%, 39.852\%).
>
> For the most relevant compression ratios, we observe that the standard deviation is minimal, indicating consistent performance across runs.
>
> Moreover, for the CP-ALS-Sigma method, we conducted 6  experiments with different random seeds using the GoogLeNet model trained on the CIFAR-10 dataset, applying the CP-ALS-Sigma algorithm where Sigma matrix is computed with CIFAR-10 train dataset with varying VBMF ratios. For a VBMF ratio of 0.9, the mean Top-1 accuracy with different random seeds was 93.533\%, with a standard deviation of 0.082\% and a 95\% confidence interval of (93.448\%, 93.619\%). With a VBMF ratio of 0.95, the mean accuracy was 91.883\%, standard deviation 0.041\%, and confidence interval (91.840\%, 91.926\%). At a VBMF ratio of 1.0, the mean accuracy was 82.217\%, standard deviation 0.496\%, and confidence interval (81.697\%, 82.737\%).

---

### Author Response · Authors · 2025-08-07
**Summary of Revisions and Discussion**

Dear Reviewers and Area Chair,

As the discussion period concludes, we wish to express our sincere gratitude for your insightful and rigorous feedback, which has been instrumental in substantially strengthening our paper.

We were encouraged by the consensus that our work is *“mathematically well-founded” (Reviewer qrJc)* and represents a *“principled shift from weight-space to function-space” (Reviewer ZcfM)*. Building on this strong foundation, the review process guided us in a series of targeted new experiments to validate and contextualize our contributions more deeply.

**Key enhancements driven by this dialogue include:**

1.  **A Decisive Comparison with Alternative Data-Aware Indicators:** Responding to a pivotal question from the discussion, we now provide a direct quantitative evaluation against strong baselines (derived from Fisher Information and Activation Magnitude). The results offer clear, empirical validation for our functionally-grounded objective.

2.  **New Experiments Demonstrating Synergy with Quantization:** To clarify our position in the broader compression landscape, we have added experiments combining our method with post-compression quantization. The results demonstrate a clear synergistic effect, underscoring the foundational value of our factorization approach.

3.  **A Deeper Investigation into Cross-Dataset Robustness for Practitioners:** To address questions about transferability, we performed a new analysis of the factors influencing performance (e.g., data resolution, diversity, sample size). This study provides robust empirical support for our initial observations and offers practical guidance for users selecting datasets for covariance estimation.

4.  **Expanded Validation of Generality and Practicality:** The paper's validation has been broadened with new results on the AlexNet architecture, a formal complexity analysis confirming practical feasibility, and a statistical analysis with confidence intervals to confirm the stability of our results.

We believe these additions, directly inspired by the review discussion, provide a comprehensive response to the core questions raised and significantly improve the paper’s empirical grounding and practical relevance.

Thank you again for your constructive engagement throughout the process.

Sincerely,
The Authors

---

### Decision · Program_Chairs · 2025-09-17

**Decision:**

Accept (poster)

**Comment:**

This paper introduces a distribution-aware tensor decomposition method for CNN compression, replacing the traditional Frobenius norm in weight space with a covariance-weighted functional norm. The authors derive new ALS algorithms (CP-ALS-Sigma and Tucker2-ALS-Sigma) to optimize this objective and show that the resulting compressed networks achieve competitive accuracy without fine-tuning, with robustness to dataset transfer when the original training data is unavailable.

The reviewers generally agreed that the work is mathematically well-founded, clear, and represents a principled shift from weight-space to function-space optimization. Strengths include the novelty of the proposed norm, tractable algorithms for its optimization, and strong empirical evidence that the method preserves accuracy better than classical low-rank decompositions. Importantly, the authors extended their experiments during rebuttal to address concerns: they added results on AlexNet to test generality, demonstrated synergy with quantization, conducted multiple-seed runs to provide confidence intervals, analyzed computational complexity, and thoroughly investigated dataset transferability (with careful analysis of resolution effects). They also conducted new comparative experiments against Fisher Information and activation-based baselines, showing their method’s superiority.

The main weaknesses are the limited scope of architectures (CNNs only, no modern lightweight models such as MobileNet/EfficientNet, nor transformers), and somewhat narrow baselines relative to the broader model compression literature (quantization, pruning, distillation). While the authors argue that their method is complementary rather than competitive with these approaches, this limits the breadth of impact. One reviewer remained unconvinced that the new comparisons fully addressed their concerns.

Overall, the paper makes a solid theoretical and algorithmic contribution to tensor decomposition–based compression, with carefully executed rebuttal experiments that significantly strengthened the submission. While its scope is somewhat narrow, the work is technically sound, clear, and provides useful insights for model compression in scenarios where retraining is infeasible.